# Adaptive Online Learning

**Dylan J. Foster** [*]
Cornell University

**Alexander Rakhlin** [†]
University of Pennsylvania

**Karthik Sridharan** [*]
Cornell University

## Abstract

We propose a general framework for studying adaptive regret bounds in the online learning setting, subsuming model selection and data-dependent bounds. Given a data- or model-dependent bound we ask, "Does there exist some algorithm achieving this bound?" We show that modifications to recently introduced sequential complexity measures can be used to answer this question by providing sufficient conditions under which adaptive rates can be achieved. In particular each adaptive rate induces a set of so-called offset complexity measures, and obtaining small upper bounds on these quantities is sufficient to demonstrate achievability. A cornerstone of our analysis technique is the use of one-sided tail inequalities to bound suprema of offset random processes.

Our framework recovers and improves a wide variety of adaptive bounds including quantile bounds, second order data-dependent bounds, and small loss bounds. In addition we derive a new type of adaptive bound for online linear optimization based on the spectral norm, as well as a new online PAC-Bayes theorem.

## 1 Introduction

Some of the recent progress on the theoretical foundations of *online learning* has been motivated by the parallel developments in the realm of *statistical learning*. In particular, this motivation has led to martingale extensions of empirical process theory, which were shown to be the "right" notions for online learnability. Two topics, however, have remained elusive thus far: obtaining data-dependent bounds and establishing model selection (or, oracle-type) inequalities for online learning problems. In this paper we develop new techniques for addressing both these questions.

Oracle inequalities and model selection have been topics of intense research in statistics in the last two decades [1, 2, 3]. Given a sequence of models $\mathcal{M}_1, \mathcal{M}_2, \ldots$ whose union is $\mathcal{M}$, one aims to derive a procedure that selects, given an i.i.d. sample of size $n$, an estimator $\hat{f}$ from a model $\mathcal{M}_{\hat{m}}$ that trades off bias and variance. Roughly speaking the desired oracle bound takes the form

$$\mathrm{err}(\hat{f}) \leq \inf_m \left\{ \inf_{f \in \mathcal{M}_m} \mathrm{err}(f) + \mathrm{pen}_n(m) \right\},$$

where $\mathrm{pen}_n(m)$ is a penalty for the model $m$. Such oracle inequalities are attractive because they can be shown to hold even if the overall model $\mathcal{M}$ is too large. A central idea in the proofs of such statements (and an idea that will appear throughout the present paper) is that $\mathrm{pen}_n(m)$ should be "slightly larger" than the fluctuations of the empirical process for the model $m$. It is therefore not surprising that concentration inequalities—and particularly Talagrand's celebrated inequality for the supremum of the empirical process—have played an important role in attaining oracle bounds. In order to select a good model in a data-driven manner, one establishes non-asymptotic data-dependent bounds on the fluctuations of an empirical process indexed by elements in each model [4].

---

[*]Deptartment of Computer Science
[†]Deptartment of Statistics

Lifting the ideas of oracle inequalities and data-dependent bounds from statistical to online learning is not an obvious task. For one, there is no concentration inequality available, even for the simple case of sequential Rademacher complexity. (For the reader already familiar with this complexity: a change of the value of one Rademacher variable results in a change of the remaining path, and hence an attempt to use a version of a bounded difference inequality grossly fails). Luckily, as we show in this paper, the concentration machinery is not needed and one only requires a one-sided tail inequality. This realization is motivated by the recent work of [5, 6, 7]. At a high level, our approach will be to develop one-sided inequalities for the suprema of certain offset processes [7], where the offset is chosen to be "slightly larger" than the complexity of the corresponding model. We then show that these offset processes determine which data-dependent adaptive rates are achievable for online learning problems, drawing strong connections to the ideas of statistical learning described earlier.

## 1.1 Framework

Let $\mathcal{X}$ be the set of observations, $\mathcal{D}$ the space of decisions, and $\mathcal{Y}$ the set of outcomes. Let $\Delta(S)$ denote the set of distributions on a set $S$. Let $\ell : \mathcal{D} \times \mathcal{Y} \to \mathbb{R}$ be a loss function. The online learning framework is defined by the following process: For $t = 1, \ldots, n$, Nature provides input instance $x_t \in \mathcal{X}$; Learner selects prediction distribution $q_t \in \Delta(\mathcal{D})$; Nature provides label $y_t \in \mathcal{Y}$, while the learner draws prediction $\hat{y}_t \sim q_t$ and suffers loss $\ell(\hat{y}_t, y_t)$.

Two important settings are *supervised learning* ($\mathcal{Y} \subseteq \mathbb{R}$, $\mathcal{D} \subseteq \mathbb{R}$) and *online linear optimization* ($\mathcal{X} = \{0\}$ is a singleton set, $\mathcal{Y}$ and $\mathcal{D}$ are balls in dual Banach spaces and $\ell(\hat{y}, y) = \langle \hat{y}, y \rangle$). For a class $\mathcal{F} \subseteq \mathcal{D}^{\mathcal{X}}$, we define the learner's cumulative *regret* to $\mathcal{F}$ as

$$\sum_{t=1}^{n} \ell(\hat{y}_t, y_t) - \inf_{f \in \mathcal{F}} \sum_{t=1}^{n} \ell(f(x_t), y_t).$$

A *uniform* regret bound $\mathcal{B}_n$ is achievable if there exists a randomized algorithm selecting $\hat{y}_t$ such that

$$\mathbb{E}\left[\sum_{t=1}^{n} \ell(\hat{y}_t, y_t) - \inf_{f \in \mathcal{F}} \sum_{t=1}^{n} \ell(f(x_t), y_t)\right] \leq \mathcal{B}_n \quad \forall x_{1:n}, y_{1:n}, \tag{1}$$

where $a_{1:n}$ stands for $\{a_1, \ldots, a_n\}$. Achievable rates $\mathcal{B}_n$ depend on complexity of the function class $\mathcal{F}$. For example, sequential Rademacher complexity of $\mathcal{F}$ is one of the tightest achievable uniform rates for a variety of loss functions [8, 7].

An *adaptive regret bound* has the form $\mathcal{B}_n(f; x_{1:n}, y_{1:n})$ and is said to be achievable if there exists a randomized algorithm for selecting $\hat{y}_t$ such that

$$\mathbb{E}\left[\sum_{t=1}^{n} \ell(\hat{y}_t, y_t) - \sum_{t=1}^{n} \ell(f(x_t), y_t)\right] \leq \mathcal{B}_n(f; x_{1:n}, y_{1:n}) \quad \forall x_{1:n}, y_{1:n}, \ \forall f \in \mathcal{F}. \tag{2}$$

We distinguish three types of adaptive bounds, according to whether $\mathcal{B}_n(f; x_{1:n}, y_{1:n})$ depends only on $f$, only on $(x_{1:n}, y_{1:n})$, or on both quantities. Whenever $\mathcal{B}_n$ depends on $f$, an adaptive regret can be viewed as an oracle inequality which penalizes each $f$ according to a measure of its complexity (e.g. the complexity of the smallest model to which it belongs). As in statistical learning, an oracle inequality (2) may be proved for certain functions $\mathcal{B}_n(f; x_{1:n}, y_{1:n})$ even if a uniform bound (1) cannot hold for any nontrivial $\mathcal{B}_n$.

## 1.2 Related Work

The case when $\mathcal{B}_n(f; x_{1:n}, y_{1:n}) = \mathcal{B}_n(x_{1:n}, y_{1:n})$ does not depend on $f$ has received most of the attention in the literature. The focus is on bounds that can be tighter for "nice sequences," yet maintain near-optimal worst-case guarantees. An incomplete list of prior work includes [9, 10, 11, 12], couched in the setting of online linear/convex optimization, and [13] in the experts setting.

A bound of type $\mathcal{B}_n(f)$ was studied in [14], which presented an algorithm that competes with all experts simultaneously, but with varied regret with respect to each of them depending on the quantile of the expert. Another bound of this type was given by [15], who consider online linear optimization with an unbounded set and provide oracle inequalities with an appropriately chosen function $\mathcal{B}_n(f)$.

Finally, the third category of adaptive bounds are those that depend on both the hypothesis $f \in \mathcal{F}$ and the data. The bounds that depend on the loss of the best function (so-called "small-loss" bounds,

[16, Sec. 2.4], [17, 13]) fall in this category trivially, since one may overbound the loss of the best function by the performance of $f$. We draw attention to the recent result of [18] who show an adaptive bound in terms of both the loss of comparator and the KL divergence between the comparator and some pre-fixed prior distribution over experts. An MDL-style bound in terms of the variance of the loss of the comparator (under the distribution induced by the algorithm) was recently given in [19].

Our study was also partly inspired by Cover [20] who characterized necessary and sufficient conditions for achievable bounds in prediction of binary sequences. The methods in [20], however, rely on the structure of the binary prediction problem and do not readily generalize to other settings.

The framework we propose recovers the vast majority of known adaptive rates in literature, including variance bounds, quantile bounds, localization-based bounds, and fast rates for small losses. It should be noted that while existing literature on adaptive online learning has focused on simple hypothesis classes such as finite experts and finite-dimensional $p$-norm balls, our results extend to general hypothesis classes, including large nonparametric ones discussed in [7].

## 2 Adaptive Rates and Achievability: General Setup

The first step in building a general theory for adaptive online learning is to identify what adaptive regret bounds are possible to achieve. Recall that an adaptive regret bound of $\mathcal{B}_n : \mathcal{F} \times \mathcal{X}^n \times \mathcal{Y}^n \to \mathbb{R}$ is said to be achievable if there exists an online learning algorithm such that, (2) holds.

In the rest of this work, we use the notation $\langle\!\langle \ldots \rangle\!\rangle_{t=1}^n$ to denote the interleaved application of the operators inside the brackets, repeated over $t = 1, \ldots, n$ rounds (see [21]). Achievability of an adaptive rate can be formalized by the following minimax quantity.

**Definition 1.** *Given an adaptive rate $\mathcal{B}_n$ we define the offset minimax value:*

$$\mathcal{A}_n(\mathcal{F}, \mathcal{B}_n) \triangleq \left\langle\!\!\left\langle \sup_{x_t \in \mathcal{X}} \inf_{q_t \in \Delta(\mathcal{D})} \sup_{y_t \in \mathcal{Y}} \mathbb{E}_{\hat{y}_t \sim q_t} \right\rangle\!\!\right\rangle_{t=1}^n \left[ \sum_{t=1}^n \ell(\hat{y}_t, y_t) - \inf_{f \in \mathcal{F}} \left\{ \sum_{t=1}^n \ell(f(x_t), y_t) + \mathcal{B}_n(f; x_{1:n}, y_{1:n}) \right\} \right].$$

$\mathcal{A}_n(\mathcal{F}, \mathcal{B}_n)$ quantifies how $\sum_{t=1}^n \ell(\hat{y}_t, y_t) - \inf_{f \in \mathcal{F}} \left\{ \sum_{t=1}^n \ell(f(x_t), y_t) + \mathcal{B}_n(f; x_{1:n}, y_{1:n}) \right\}$ behaves when the optimal learning algorithm that minimizes this difference is used against Nature trying to maximize it. Directly from this definition,

$$\text{An adaptive rate } \mathcal{B}_n \text{ is achievable if and only if } \mathcal{A}_n(\mathcal{F}, \mathcal{B}_n) \leq 0.$$

If $\mathcal{B}_n$ is a uniform rate, i.e., $\mathcal{B}_n(f; x_{1:n}, y_{1:n}) = \mathcal{B}_n$, achievability reduces to the minimax analysis explored in [8]. The uniform rate $\mathcal{B}_n$ is achievable if and only if $\mathcal{B}_n \geq \mathcal{V}_n(\mathcal{F})$, where $\mathcal{V}_n(\mathcal{F})$ is the minimax value of the online learning game.

We now focus on understanding the minimax value $\mathcal{A}_n(\mathcal{F}, \mathcal{B}_n)$ for general adaptive rates. We first show that the minimax value is bounded by an offset version of the sequential Rademacher complexity studied in [8]. The symmetrization Lemma 1 below provides us with the first step towards a probabilistic analysis of achievable rates. Before stating the lemma, we need to define the notion of a tree and the notion of sequential Rademacher complexity.

Given a set $\mathcal{Z}$, a $\mathcal{Z}$-valued tree $\mathbf{z}$ of depth $n$ is a sequence $(\mathbf{z}_t)_{t=1}^n$ of functions $\mathbf{z}_t : \{\pm 1\}^{t-1} \to \mathcal{Z}$. One may view $\mathbf{z}$ as a complete binary tree decorated by elements of $\mathcal{Z}$. Let $\epsilon = (\epsilon_t)_{t=1}^n$ be a sequence of independent Rademacher random variables. Then $(\mathbf{z}_t(\epsilon))$ may be viewed as a predictable process with respect to the filtration $\mathcal{S}_t = \sigma(\epsilon_1, \ldots, \epsilon_t)$. For a tree $\mathbf{z}$, the sequential Rademacher complexity of a function class $\mathcal{G} \subseteq \mathbb{R}^{\mathcal{Z}}$ on $\mathbf{z}$ is defined as

$$\mathcal{R}_n(\mathcal{G}, \mathbf{z}) \triangleq \mathbb{E}_\epsilon \sup_{g \in \mathcal{G}} \sum_{t=1}^n \epsilon_t g(\mathbf{z}_t(\epsilon)) \quad \text{and} \quad \mathcal{R}_n(\mathcal{G}) \triangleq \sup_{\mathbf{z}} \mathcal{R}_n(\mathcal{G}, \mathbf{z}).$$

**Lemma 1.** *For any lower semi-continuous loss $\ell$, and any adaptive rate $\mathcal{B}_n$ that only depends on outcomes (i.e. $\mathcal{B}_n(f; x_{1:n}, y_{1:n}) = \mathcal{B}_n(y_{1:n})$), we have that*

$$\mathcal{A}_n \leq \sup_{\mathbf{x}, \mathbf{y}} \mathbb{E}_\epsilon \left[ \sup_{f \in \mathcal{F}} \left\{ 2 \sum_{t=1}^n \epsilon_t \ell(f(\mathbf{x}_t(\epsilon)), \mathbf{y}_t(\epsilon)) \right\} - \mathcal{B}_n(\mathbf{y}_{1:n}(\epsilon)) \right]. \tag{3}$$

*Further, for any general adaptive rate $\mathcal{B}_n$,*

$$\mathcal{A}_n \le \sup_{\mathbf{x},\mathbf{y},\mathbf{y}'} \mathbb{E}_\epsilon \left[ \sup_{f \in \mathcal{F}} \left\{ 2 \sum_{t=1}^n \epsilon_t \ell(f(\mathbf{x}_t(\epsilon)), \mathbf{y}_t(\epsilon)) - \mathcal{B}_n(f; \mathbf{x}_{1:n}(\epsilon), \mathbf{y}'_{2:n+1}(\epsilon)) \right\} \right]. \tag{4}$$

*Finally, if one considers the supervised learning problem where $\mathcal{F}: \mathcal{X} \to \mathbb{R}$, $\mathcal{Y} \subset \mathbb{R}$ and $\ell: \mathbb{R} \times \mathbb{R} \to \mathbb{R}$ is a loss that is convex and L-Lipschitz in its first argument, then for any adaptive rate $\mathcal{B}_n$,*

$$\mathcal{A}_n \le \sup_{\mathbf{x},\mathbf{y}} \mathbb{E}_\epsilon \left[ \sup_{f \in \mathcal{F}} \left\{ 2L \sum_{t=1}^n \epsilon_t f(\mathbf{x}_t(\epsilon)) - \mathcal{B}_n(f; \mathbf{x}_{1:n}(\epsilon), \mathbf{y}_{1:n}(\epsilon)) \right\} \right]. \tag{5}$$

The above lemma tells us that to check whether an adaptive rate is achievable, it is sufficient to check that the corresponding adaptive sequential complexity measures are non-positive. We remark that if the above complexities are bounded by some positive quantity of a smaller order, one can form a new achievable rate $\mathcal{B}'_n$ by adding the positive quantity to $\mathcal{B}_n$.

## 3 Probabilistic Tools

As mentioned in the introduction, our technique rests on certain one-sided probabilistic inequalities. We now state the first building block: a rather straightforward maximal inequality.

**Proposition 2.** *Let $I = \{1, \dots, N\}$, $N \le \infty$, be a set of indices and let $(X_i)_{i \in I}$ be a sequence of random variables satisfying the following tail condition: for any $\tau > 0$,*

$$P(X_i - B_i > \tau) \le C_1 \exp\left(-\tau^2/(2\sigma_i^2)\right) + C_2 \exp\left(-\tau s_i\right) \tag{6}$$

*for some positive sequence $(B_i)$, nonnegative sequence $(\sigma_i)$ and nonnegative sequence $(s_i)$ of numbers, and for constants $C_1, C_2 \ge 0$. Then for any $\bar{\sigma} \le \sigma_1$, $\bar{s} \ge s_1$, and*

$$\theta_i = \max\left\{ \frac{\sigma_i}{B_i} \sqrt{2\log(\sigma_i/\bar{\sigma}) + 4\log(i)}, (B_i s_i)^{-1} \log\left(i^2(\bar{s}/s_i)\right) \right\} + 1,$$

*it holds that*
$$\mathbb{E} \sup_{i \in I} \left\{ X_i - B_i \theta_i \right\} \le 3 C_1 \bar{\sigma} + 2 C_2 (\bar{s})^{-1}. \tag{7}$$

We remark that $B_i$ need not be the expected value of $X_i$, as we are not interested in two-sided deviations around the mean.

One of the approaches to obtaining oracle-type inequalities is to split a large class into smaller ones according to a "complexity radius" and control a certain stochastic process separately on each subset (also known as the *peeling* technique). In the applications below, $X_i$ will often stand for the (random) supremum of this process on subset $i$, and $B_i$ will be an upper bound on its typical size. Given deviation bounds for $X_i$ above $B_i$, the dilated size $B_i \theta_i$ then allows one to pass to maximal inequalities (7) and thus verify achievability in Lemma 1. The same strategy works for obtaining data-dependent bounds, where we first prove tail bounds for the given size of the data-dependent quantity, then appeal to (7).

A simple yet powerful example for the control of the supremum of a stochastic process is an inequality due to Pinelis [22] for the norm (which is a supremum over the dual ball) of a martingale in a 2-smooth Banach space. Here we state a version of this result that can be found in [23, Appendix A].

**Lemma 3.** *Let $\mathcal{Z}$ be a unit ball in a separable $(2, D)$-smooth Banach space $\mathcal{H}$. For any $\mathcal{Z}$-valued tree $\mathbf{z}$, and any $n > \tau/4D^2$*

$$P\left( \left\| \sum_{t=1}^n \epsilon_t \mathbf{z}_t(\epsilon) \right\| \ge \tau \right) \le 2 \exp\left( -\frac{\tau^2}{8D^2 n} \right)$$

When the class of functions is not linear, we may no longer appeal to the above lemma. Instead, we make use of a result from [24] that extends Lemma 3 at a price of a poly-logarithmic factor. Before stating this lemma, we briefly define the relevant complexity measures (see [24] for more details). First, a set $V$ of $\mathbb{R}$-valued trees is called an $\alpha$-cover of $\mathcal{G} \subseteq \mathbb{R}^{\mathcal{Z}}$ on $\mathbf{z}$ with respect to $\ell_p$ if

$$\forall g \in \mathcal{G}, \forall \epsilon \in \{\pm 1\}^n, \exists \mathbf{v} \in V \quad \text{s.t.} \quad \sum_{t=1}^n (g(\mathbf{z}_t(\epsilon)) - \mathbf{v}_t(\epsilon))^p \le n\alpha^p.$$

The size of the smallest $\alpha$-cover is denoted by $\mathcal{N}_p(\mathcal{G}, \alpha, \mathbf{z})$, and $\mathcal{N}_p(\mathcal{G}, \alpha, n) \triangleq \sup_{\mathbf{z}} \mathcal{N}_p(\mathcal{G}, \alpha, \mathbf{z})$.

The set $V$ is an $\alpha$-cover of $\mathcal{G}$ on $\mathbf{z}$ with respect to $\ell_\infty$ if

$$\forall g \in \mathcal{G}, \forall \epsilon \in \{\pm 1\}, \exists \mathbf{v} \in V \quad \text{s.t.} \quad |g(\mathbf{z}_t(\epsilon)) - \mathbf{v}_t(\epsilon)| \le \alpha \quad \forall t \in [n].$$

We let $\mathcal{N}_\infty(\mathcal{G}, \alpha, \mathbf{z})$ be the smallest such cover and set $\mathcal{N}_\infty(\mathcal{G}, \alpha, n) = \sup_{\mathbf{z}} \mathcal{N}_\infty(\mathcal{G}, \alpha, \mathbf{z})$.

**Lemma 4** ([24]). *Let $\mathcal{G} \subseteq [-1,1]^{\mathcal{Z}}$. Suppose $\mathcal{R}_n(\mathcal{G})/n \to 0$ with $n \to \infty$ and that the following mild assumptions hold: $\mathcal{R}_n(\mathcal{G}) \ge 1/n$, $\mathcal{N}_\infty(\mathcal{G}, 2^{-1}, n) \ge 4$, and there exists a constant $\Gamma$ such that $\Gamma \ge \sum_{j=1}^{\infty} \mathcal{N}_\infty(\mathcal{G}, 2^{-j}, n)^{-1}$. Then for any $\theta > \sqrt{12/n}$, for any $\mathcal{Z}$-valued tree $\mathbf{z}$ of depth $n$,*

$$P\left(\sup_{g \in \mathcal{G}} \left|\sum_{t=1}^{n} \epsilon_t g(\mathbf{z}_t(\epsilon))\right| > 8\left(1 + \theta\sqrt{8n \log^3(en^2)}\right) \cdot \mathcal{R}_n(\mathcal{G})\right)$$

$$\le P\left(\sup_{g \in \mathcal{G}} \left|\sum_{t=1}^{n} \epsilon_t g(\mathbf{z}_t(\epsilon))\right| > n \inf_{\alpha > 0}\left\{4\alpha + 6\theta \int_\alpha^1 \sqrt{\log \mathcal{N}_\infty(\mathcal{G}, \delta, n)} d\delta\right\}\right) \le 2\Gamma e^{-\frac{n\theta^2}{4}}.$$

The above lemma yields a one-sided control on the size of the supremum of the sequential Rademacher process, as required for our oracle-type inequalities.

Next, we turn our attention to an offset Rademacher process, where the supremum is taken over a collection of negative-mean random variables. The behavior of this offset process was shown to govern the optimal rates of convergence for online nonparametric regression [7]. Such a one-sided control of the supremum will be necessary for some of the data-dependent upper bounds we develop.

**Lemma 5.** *Let $\mathbf{z}$ be a $\mathcal{Z}$-valued tree of depth $n$, and let $\mathcal{G} \subseteq \mathbb{R}^{\mathcal{Z}}$. For any $\gamma \ge 1/n$ and $\alpha > 0$,*

$$P\left(\sup_{g \in \mathcal{G}} \sum_{t=1}^{n} \left(\epsilon_t g(\mathbf{z}_t(\epsilon)) - 2\alpha g^2(\mathbf{z}_t(\epsilon))\right) - \frac{\log \mathcal{N}_2(\mathcal{G}, \gamma, \mathbf{z})}{\alpha} - 12\sqrt{2}\int_{1/n}^{\gamma} \sqrt{n \log \mathcal{N}_2(\mathcal{G}, \delta, \mathbf{z})} d\delta - 1 > \tau\right)$$

$$\le \Gamma \exp\left(-\frac{\tau^2}{2\sigma^2}\right) + \exp\left(-\frac{\alpha\tau}{2}\right),$$

*where $\Gamma \ge \sum_{j=1}^{\log_2(2n\gamma)} \mathcal{N}_2(\mathcal{G}, 2^{-j}\gamma, \mathbf{z})^{-2}$ and $\sigma = 12\int_{\frac{1}{n}}^{\gamma} \sqrt{n \log \mathcal{N}_2(\mathcal{G}, \delta, \mathbf{z})} d\delta$.*

We observe that the probability of deviation has both subgaussian and subexponential components.

Using the above result and Proposition 2 leads to useful bounds on the quantities in Lemma 1 for specific types of adaptive rates. Given a tree $\mathbf{z}$, we obtain a bound on the expected size of the sequential Rademacher process when we subtract off the data-dependent $\ell_2$-norm of the function on the tree $\mathbf{z}$, adjusted by logarithmic terms.

**Corollary 6.** *Suppose $\mathcal{G} \subseteq [-1,1]^{\mathcal{Z}}$, and let $\mathbf{z}$ be any $\mathcal{Z}$-valued tree of depth $n$. Assume $\log \mathcal{N}_2(\mathcal{G}, \delta, n) \le \delta^{-p}$ for some $p < 2$. Then*

$$\mathbb{E} \sup_{g \in \mathcal{G}, \gamma}\left\{\sum_{t=1}^{n} \epsilon_t g(\mathbf{z}_t(\epsilon)) - 4\sqrt{2(\log n) \log \mathcal{N}_2(\mathcal{G}, \gamma/2, \mathbf{z}) \left(\sum_{t=1}^{n} g^2(\mathbf{z}_t(\epsilon)) + 1\right)}\right.$$

$$\left. -24\sqrt{2} \log n \int_{1/n}^{\gamma} \sqrt{n \log \mathcal{N}_2(\mathcal{G}, \delta, \mathbf{z})} d\delta\right\} \le 7 + 2\log n.$$

The next corollary yields slightly faster rates than Corollary 6 when $|\mathcal{G}| < \infty$.

**Corollary 7.** *Suppose $\mathcal{G} \subseteq [-1,1]^{\mathcal{Z}}$ with $|\mathcal{G}| = N$, and let $\mathbf{z}$ be any $\mathcal{Z}$-valued tree of depth $n$. Then*

$$\mathbb{E} \sup_{g \in \mathcal{G}}\left\{\sum_{t=1}^{n} \epsilon_t g(\mathbf{z}_t(\epsilon)) - 2\log\left(\log N \sum_{t=1}^{n} g^2(\mathbf{z}(\epsilon)) + e\right) \sqrt{32\left(\log N \sum_{t=1}^{n} g^2(\mathbf{z}(\epsilon)) + e\right)}\right\} \le 1.$$

## 4 Achievable Bounds

In this section we use Lemma 1 along with the probabilistic tools from the previous section to obtain an array of achievable adaptive bounds for various online learning problems. We subdivide the section into one subsection for each category of adaptive bound described in Section 1.1.

## 4.1 Adapting to Data

Here we consider adaptive rates of the form $\mathcal{B}_n(x_{1:n}, y_{1:n})$, uniform over $f \in \mathcal{F}$. We show the power of the developed tools on the following example.

**Example 4.1** (**Online Linear Optimization in** $\mathbb{R}^d$). *Consider the problem of online linear optimization where $\mathcal{F} = \{f \in \mathbb{R}^d : \|f\|_2 \leq 1\}$, $\mathcal{Y} = \{y : \|y\|_2 \leq 1\}$, $\mathcal{X} = \{0\}$, and $\ell(\hat{y}, y) = \langle \hat{y}, y \rangle$. The following adaptive rate is achievable:*

$$\mathcal{B}_n(y_{1:n}) = 16\sqrt{d}\log(n)\left\|\left(\textstyle\sum_{t=1}^n y_t y_t^\top\right)^{1/2}\right\|_\sigma + 16\sqrt{d}\log(n),$$

*where $\|\cdot\|_\sigma$ is the spectral norm. Let us deduce this result from Corollary 6. First, observe that*

$$\left\|\left(\textstyle\sum_{t=1}^n y_t y_t^\top\right)^{1/2}\right\|_\sigma = \sup_{f:\|f\|_2 \leq 1}\left\|\left(\textstyle\sum_{t=1}^n y_t y_t^\top\right)^{1/2} f\right\| = \sup_{f:\|f\|_2 \leq 1}\sqrt{f^\top \textstyle\sum_{t=1}^n y_t y_t^\top f} = \sup_{f \in \mathcal{F}}\sqrt{\textstyle\sum_{t=1}^n \ell^2(f, y_t)}.$$

*The linear function class $\mathcal{F}$ can be covered point-wise at any scale $\delta$ with $(3/\delta)^d$ balls and thus $\mathcal{N}(\ell \circ \mathcal{F}, 1/(2n), \mathbf{z}) \leq (6n)^d$ for any $\mathcal{Y}$-valued tree $\mathbf{z}$. We apply Corollary 6 with $\gamma = 1/n$ and the integral term in the corollary vanishes, yielding the claimed statement.*

## 4.2 Model Adaptation

In this subsection we focus on achievable rates for oracle inequalities and model selection, but without dependence on data. The form of the rate is therefore $\mathcal{B}_n(f)$. Assume we have a class $\mathcal{F} = \bigcup_{R \geq 1} \mathcal{F}(R)$, with the property that $\mathcal{F}(R) \subseteq \mathcal{F}(R')$ for any $R \leq R'$. If we are told by an oracle that regret will be measured with respect to those hypotheses $f \in \mathcal{F}$ with $R(f) \triangleq \inf\{R : f \in \mathcal{F}(R)\} \leq R^*$, then using the minimax algorithm one can guarantee a regret bound of at most the sequential Rademacher complexity $\mathcal{R}_n(\mathcal{F}(R^*))$. On the other hand, given the optimality of the sequential Rademacher complexity for online learning problems for commonly encountered losses, we can argue that for any $f \in \mathcal{F}$ chosen in hindsight, one cannot expect a regret better than order $\mathcal{R}_n(\mathcal{F}(R(f)))$. In this section we show that simultaneously for all $f \in \mathcal{F}$, one can attain an adaptive upper bound of $O\left(\mathcal{R}_n(\mathcal{F}(R(f)))\sqrt{\log\left(\mathcal{R}_n(\mathcal{F}(R(f)))\right)}\log^{3/2} n\right)$. That is, we may predict as if we knew the optimal radius, at the price of a logarithmic factor. This is the price of adaptation.

**Corollary 8.** *For any class of predictors $\mathcal{F}$ with $\mathcal{F}(1)$ non-empty, if one considers the supervised learning problem with 1-Lipschitz loss $\ell$, the following rate is achievable:*

$$\mathcal{B}_n(f) = \log^{3/2} n\left(K_1\mathcal{R}_n(\mathcal{F}(2R(f)))\left(1 + \sqrt{\log\left(\frac{\log(2R(f)) \cdot \mathcal{R}_n(\mathcal{F}(2R(f)))}{\mathcal{R}_n(\mathcal{F}(1))}\right)}\right) + K_2\Gamma\mathcal{R}_n(\mathcal{F}(1))\right),$$

*for absolute constants $K_1, K_2$, and $\Gamma$ defined in Lemma 4.*

In fact, this statement is true more generally with $\mathcal{F}(2R(f))$ replaced by $\ell \circ \mathcal{F}(2R(f))$. It is tempting to attempt to prove the above statement with the exponential weights algorithm running as an aggregation procedure over the solutions for each $R$. In general, this approach will fail for two reasons. First, if function values grow with $R$, the exponential weights bound will scale linearly with this value. Second, an experts bound yields only a slower $\sqrt{n}$ rate.

As a special case of the above lemma, we obtain an *online PAC-Bayesian theorem*. We postpone this example to the next sub-section where we get a *data-dependent* version of this result. We now provide a bound for online linear optimization in 2-smooth Banach spaces that automatically adapts to the norm of the comparator. To prove it, we use the concentration bound from [22] (Lemma 3) within the proof of the above corollary to remove the extra logarithmic factors.

**Example 4.2** (**Unconstrained Linear Optimization**). *Consider linear optimization with $\mathcal{Y}$ being the unit ball of some reflexive Banach space with norm $\|\cdot\|_*$. Let $\mathcal{F} = \mathcal{D}$ be the dual space and the loss $\ell(\hat{y}, y) = \langle \hat{y}, y \rangle$ (where we are using $\langle \cdot, \cdot \rangle$ to represent the linear functional in the first argument to the second argument). Define $\mathcal{F}(R) = \{f \mid \|f\| \leq R\}$ where $\|\cdot\|$ is the norm dual to $\|\cdot\|_*$. If the unit ball of $\mathcal{Y}$ is $(2, D)$-smooth, then the following rate is achievable for all $f$ with $\|f\| \geq 1$:*

$$\mathcal{B}(f) = D\sqrt{n}\left(8\|f\|\left(1 + \sqrt{\log(2\|f\|) + \log\log(2\|f\|)}\right) + 12\right).$$

*For the case of a Hilbert space, the above bound was achieved by [15].*

### 4.3 Adapting to Data and Model Simultaneously

We now study achievable bounds that perform online model selection in a data-adaptive way. Of specific interest is our online optimistic PAC-Bayesian bound. This bound should be compared to [18, 19], with the reader noting that it is independent of the number of experts, is algorithm-independent, and depends quadratically on the expected loss of the expert we compare against.

**Example 4.3** (**Generalized Predictable Sequences (Supervised Learning)**). *Consider an online supervised learning problem with a convex 1-Lipschitz loss. Let $(M_t)_{t\geq 1}$ be any predictable sequence that the learner can compute at round $t$ based on information provided so far, including $x_t$ (One can think of the predictable sequence $M_t$ as a prior guess for the hypothesis we would compare with in hindsight). Then the following adaptive rate is achievable:*

$$\mathcal{B}_n(f; x_{1:n}) = \inf_\gamma \left\{ K_1 \sqrt{\log n \cdot \log \mathcal{N}_2(\mathcal{F}, \gamma/2, n) \cdot \left( \sum_{t=1}^n (f(x_t) - M_t)^2 + 1 \right)} \right.$$
$$\left. + K_2 \log n \int_{1/n}^\gamma \sqrt{n \log \mathcal{N}_2(\mathcal{F}, \delta, n)} d\delta + 2 \log n + 7 \right\},$$

*for constants $K_1 = 4\sqrt{2}, K_2 = 24\sqrt{2}$ from Corollary 6. The achievability is a direct consequence of Eq. (5) in Lemma 1, followed by Corollary 6 (one can include any predictable sequence in the Rademacher average part because $\sum_t M_t \epsilon_t$ is zero mean). Particularly, if we assume that the sequential covering of class $\mathcal{F}$ grows as $\log \mathcal{N}_2(\mathcal{F}, \epsilon, n) \leq \epsilon^{-p}$ for some $p < 2$, we get that*

$$\mathcal{B}_n(f) = \tilde{O}\left( \left( \sqrt{\sum_{t=1}^n (f(x_t) - M_t)^2 + 1} \right)^{1-\frac{p}{2}} (\sqrt{n})^{p/2} \right).$$

*As $p$ gets closer to 0, we get full adaptivity and replace $n$ by $\sum_{t=1}^n (f(x_t) - M_t)^2 + 1$. On the other hand, as $p$ gets closer to 2 (i.e. more complex function classes), we do not adapt and get a uniform bound in terms of $n$. For $p \in (0, 2)$, we attain a natural interpolation.*

**Example 4.4** (**Regret to Fixed Vs Regret to Best (Supervised Learning)**). *Consider an online supervised learning problem with a convex 1-Lipschitz loss and let $|\mathcal{F}| = N$. Let $f^\star \in \mathcal{F}$ be a fixed expert chosen in advance. The following bound is achievable:*

$$\mathcal{B}_n(f, x_{1:n}) = 4 \log \left( \log N \sum_{t=1}^n (f(x_t) - f^\star(x_t))^2 + e \right) \sqrt{32 \left( \log N \sum_{t=1}^n (f(x_t) - f^\star(x_t))^2 + e \right)} + 2.$$

*In particular, against $f^\star$ we have $\mathcal{B}_n(f^\star, x_{1:n}) = O(1)$, and against an arbitrary expert we have $\mathcal{B}_n(f, x_{1:n}) = O(\sqrt{n \log N}(\log(n \cdot \log N)))$. This bound follows from Eq. (5) in Lemma 1 followed by Corollary 7. This extends the study of [25] to supervised learning and general class of experts $\mathcal{F}$.*

**Example 4.5** (**Optimistic PAC-Bayes**). *Assume that we have a countable set of experts and that the loss for each expert on any round is non-negative and bounded by 1. The function class $\mathcal{F}$ is the set of all distributions over these experts, and $\mathcal{X} = \{0\}$. This setting can be formulated as online linear optimization where the loss of mixture $f$ over experts, given instance $y$, is $\langle f, y \rangle$, the expected loss under the mixture. The following adaptive bound is achievable:*

$$\mathcal{B}_n(f; y_{1:n}) = \sqrt{50 \left( \mathrm{KL}(f|\pi) + \log(n) \right) \sum_{t=1}^n \mathbb{E}_{i \sim f} \langle e_i, y_t \rangle^2 + 50 \left( \mathrm{KL}(f|\pi) + \log(n) \right) + 10}.$$

*This adaptive bound is an **online PAC-Bayesian bound**. The rate adapts not only to the KL divergence of $f$ with fixed prior $\pi$ but also replaces $n$ with $\sum_{t=1}^n \mathbb{E}_{i \sim f} \langle e_i, y_t \rangle^2$. Note that we have $\sum_{t=1}^n \mathbb{E}_{i \sim f} \langle e_i, y_t \rangle^2 \leq \sum_{t=1}^n \langle f, y_t \rangle$, yielding the small-loss type bound described earlier. This is an improvement over the bound in [18] in that the bound is independent of number of experts, and so holds even for countably infinite sets of experts. The KL term in our bound may be compared to the MDL-style term in the bound of [19]. If we have a large (but finite) number of experts and take $\pi$ to be uniform, the above bound provides an improvement over both [14][1] and [18].*

*Evaluating the above bound with a distribution $f$ that places all its weight on any one expert appears to address the open question posed by [13] of obtaining algorithm-independent oracle-type variance bounds for experts. The proof of achievability of the above rate is shown in the appendix because it requires a slight variation on the symmetrization lemma specific to the problem.*

# 5 Relaxations for Adaptive Learning

To design algorithms for achievable rates, we extend the framework of online relaxations from [26]. A relaxation $\mathbf{Rel}_n : \bigcup_{t=0}^n \mathcal{X}^t \times \mathcal{Y}^t \to \mathbb{R}$ that satisfies the *initial condition*,

$$\mathbf{Rel}_n(x_{1:n}, y_{1:n}) \geq - \inf_{f \in \mathcal{F}} \left\{ \sum_{t=1}^n \ell(f(x_t), y_t) + \mathcal{B}_n(f; x_{1:n}, y_{1:n}) \right\}, \tag{8}$$

and the *recursive condition*,

$$\mathbf{Rel}_n(x_{1:t-1}, y_{1:t-1}) \geq \sup_{x_t \in \mathcal{X}} \inf_{q_t \in \Delta(\mathcal{D})} \sup_{y_t \in \mathcal{Y}} \mathbb{E}_{\hat{y} \sim q_t}[\ell(\hat{y}_t, y_t) + \mathbf{Rel}_n(x_{1:t}, y_{1:t})], \tag{9}$$

is said to be *admissible for the adaptive rate* $\mathcal{B}_n$. The relaxation's corresponding strategy is $\hat{q}_t = \arg\min_{q_t \in \Delta(\mathcal{D})} \sup_{y_t \in \mathcal{Y}} \mathbb{E}_{\hat{y} \sim q_t}[\ell(\hat{y}_t, y_t) + \mathbf{Rel}_n(x_{1:t}, y_{1:t})]$, which enjoys the adaptive bound

$$\sum_{t=1}^n \ell(\hat{y}_t, y_t) - \inf_{f \in \mathcal{F}} \left\{ \sum_{t=1}^n \ell(f(x_t), y_t) + \mathcal{B}_n(f; x_{1:n}, y_{1:n}) \right\} \leq \mathbf{Rel}_n(\cdot) \quad \forall x_{1:n}, y_{1:n}.$$

It follows immediately that the strategy achieves the rate $\mathcal{B}_n(f; x_{1:n}, y_{1:n}) + \mathbf{Rel}_n(\cdot)$. Our goal is then to find relaxations for which the strategy is computationally tractable and $\mathbf{Rel}_n(\cdot) \leq 0$ or at least has smaller order than $\mathcal{B}_n$. Similar to [26], conditional versions of the offset minimax values $\mathcal{A}_n$ yield admissible relaxations, but solving these relaxations may not be computationally tractable.

**Example 5.1 (Online PAC-Bayes).** *Consider the experts setting in Example 4.5 with:*

$$\mathcal{B}_n(f) = 3\sqrt{2n \max\{KL(f \mid \pi), 1\}} + 4\sqrt{n}.$$

*Let $R_i = 2^{i-1}$ and let $q_t^R(y)$ denote the exponential weights distribution with learning rate $\sqrt{R/n}$:* $q^R(y_{1:t})_k \propto \pi_k \exp\left(-\sqrt{R/n}(\sum_{s=1}^t y_t)_k\right)$. *The following is an admissible relaxation achieving $\mathcal{B}_n$:*

$$\mathbf{Rel}_n(y_{1:t}) = \inf_{\lambda > 0} \left[ \frac{1}{\lambda} \log \left( \sum_i \exp\left( -\lambda \left[ \sum_{s=1}^t \langle q^{R_i}(y_{1:s-1}), y_s \rangle + \sqrt{nR_i} \right] \right) \right) + 2\lambda(n-t) \right].$$

*Let $q_t^\star$ be a distribution with $(q_t^\star)_i \propto \exp\left( -\frac{1}{\sqrt{n}}\left[ \sum_{s=1}^{t-1} \langle q^{R_i}(y_{1:s-1}), y_s \rangle - \sqrt{nR_i} \right] \right)$. We predict by drawing $i$ according to $q_t^\star$, then drawing an expert according to $q^{R_i}(y_{1:t-1})$.*

While in general the problem of obtaining an efficient adaptive relaxation might be hard, one can ask the question, "If and efficient relaxation $\mathbf{Rel}_n^R$ is available for each $\mathcal{F}(R)$, can one obtain an adaptive model selection algorithm for all of $\mathcal{F}$?". To this end for supervised learning problem with convex Lipschitz loss we delineate a meta approach which utilizes existing relaxations for each $\mathcal{F}(R)$.

**Lemma 9.** *Let $q_t^R(y_1, \ldots, y_{t-1})$ be the randomized strategy corresponding to $\mathbf{Rel}_n^R$, obtained after observing outcomes $y_1, \ldots, y_{t-1}$, and let $\theta : \mathbb{R} \to \mathbb{R}$ be nonnegative. The following relaxation is admissible for the rate $\mathcal{B}_n(R) = \mathbf{Rel}_n^R(\cdot)\theta(\mathbf{Rel}_n^R(\cdot))$:*

$$\mathbf{Ada}_n(x_{1:t}, y_{1:t}) =$$

$$\sup_{\mathbf{x}, \mathbf{y}, \mathbf{y}'} \mathbb{E}_{\epsilon_{t+1:n}} \sup_{R \geq 1} \left[ \mathbf{Rel}_n^R(x_{1:t}, y_{1:t}) - \mathbf{Rel}_n^R(\cdot)\theta(\mathbf{Rel}_n^R(\cdot)) + 2 \sum_{s=t+1}^n \epsilon_s \mathbb{E}_{\hat{y}_s \sim q_s^R(y_{1:t}, \mathbf{y}'_{t+1:s-1}(\epsilon))} \ell(\hat{y}_s, \mathbf{y}_s(\epsilon)) \right].$$

Playing according to the strategy for $\mathbf{Ada}_n$ will guarantee a regret bound of $\mathcal{B}_n(R) + \mathbf{Ada}_n(\cdot)$, and $\mathbf{Ada}_n(\cdot)$ can be bounded using Proposition 2 when the form of $\theta$ is as in that proposition.

We remark that the above strategy is not necessarily obtained by running a high-level experts algorithm over the discretized values of $R$. It is an interesting question to determine the cases when such a strategy is optimal. More generally, when the adaptive rate $\mathcal{B}_n$ depends on data, it is not possible to obtain the rates we show non-constructively in this paper using the exponential weights algorithm with meta-experts as the required weighting over experts would be data dependent (and hence is not a prior over experts). Further, the bounds from exponential-weights-type algorithms are akin to having sub-exponential tails in Proposition 2, but for many problems we may have sub-gaussian tails.

Obtaining computationally efficient methods from the proposed framework is an interesting research direction. Proposition 2 provides a useful non-constructive tool to establish achievable adaptive bounds, and a natural question to ask is if one can obtain a constructive counterpart for the proposition.

## Footnotes

[1] See [18] for a comparison of KL-based bounds and quantile bounds.

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
