[Supplementary Material]

# A   Adaptive Rates and Achievability

***Proof of Lemma 1.***  We first prove Eq. (3) and (4). We start from the definition of $\mathcal{A}_n(\mathcal{F})$. Our proof proceeds "inside out" by starting with the $n^{th}$ term and then working backwards by repeatedly applying the minimax theorem. To this end on similar lines as in [24, 7, 21], we start with the inner most term as,

$$
\sup_{x_n \in \mathcal{X}} \inf_{q_n \in \Delta(\mathcal{D})} \sup_{y_n \in \mathcal{Y}} \left( \mathbb{E}_{\hat{y}_n \sim q_n} \left[ \ell(\hat{y}_n, y_n) - \inf_{f \in \mathcal{F}} \left\{ \sum_{t=1}^{n} \ell(f(x_t), y_t) + \mathcal{B}_n(f; x_{1:n}, y_{1:n}) \right\} \right] \right)
$$

$$
= \sup_{x_n \in \mathcal{X}} \inf_{q_n \in \Delta(\mathcal{D})} \sup_{p_n \in \Delta(\mathcal{Y})} \left( \mathbb{E}_{\substack{\hat{y}_n \sim q_n \\ y_n \sim p_n}} \left[ \sum_{t=1}^{n} \ell(\hat{y}_t, y_t) - \inf_{f \in \mathcal{F}} \left\{ \sum_{t=1}^{n} \ell(f(x_t), y_t) + \mathcal{B}_n(f; x_{1:n}, y_{1:n}) \right\} \right] \right)
$$

$$
= \sup_{x_n \in \mathcal{X}} \sup_{p_n \in \Delta(\mathcal{Y})} \inf_{q_n \in \Delta(\mathcal{D})} \left( \mathbb{E}_{\substack{\hat{y}_n \sim q_n \\ y_n \sim p_n}} \left[ \sum_{t=1}^{n} \ell(\hat{y}_t, y_t) - \inf_{f \in \mathcal{F}} \left\{ \sum_{t=1}^{n} \ell(f(x_t), y_t) + \mathcal{B}_n(f; x_{1:n}, y_{1:n}) \right\} \right] \right)
$$

$$
= \sup_{x_n \in \mathcal{X}} \sup_{p_n \in \Delta(\mathcal{Y})} \inf_{\hat{y}_n \in \mathcal{D}} \left( \mathbb{E}_{y_n \sim p_n} \left[ \sum_{t=1}^{n} \ell(\hat{y}_t, y_t) - \inf_{f \in \mathcal{F}} \left\{ \sum_{t=1}^{n} \ell(f(x_t), y_t) + \mathcal{B}_n(f; x_{1:n}, y_{1:n}) \right\} \right] \right)
$$

$$
= \sup_{x_n \in \mathcal{X}} \sup_{p_n \in \Delta(\mathcal{Y})} \left( \mathbb{E}_{y_n \sim p_n} \left[ \sup_{f \in \mathcal{F}} \left\{ \inf_{\hat{y}_n \in \mathcal{D}} \mathbb{E}_{y_n \sim p_n} \left[ \sum_{t=1}^{n} \ell(\hat{y}_t, y_t) \right] - \sum_{t=1}^{n} \ell(f(x_t), y_t) - \mathcal{B}_n(f; x_{1:n}, y_{1:n}) \right\} \right] \right).
$$

To apply the minimax theorem in step 3 above, we note that the term in the round bracket is linear in $q_n$ and in $p_n$ (as it is an expectation). Hence under mild assumptions on the sets $\mathcal{D}$ and $\mathcal{Y}$, the losses, and the adaptive rate $\mathcal{B}_n$, one can apply a generalized version of the minimax theorem to swap $\sup_{p_n}$ and $\inf_{q_n}$. Compactness of the sets and lower semi-continuity of the losses and $\mathcal{B}_n$ are sufficient, but see [24, 21] for milder conditions. Proceeding backward from $n$ to 1 in a similar fashion we end up with the following quantity:

$$
\mathcal{A}_n(\mathcal{F})
$$

$$
= \left\langle\!\!\left\langle \sup_{x_t \in \mathcal{X}} \inf_{q_t \in \Delta(\mathcal{D})} \sup_{y_t \in \mathcal{Y}} \mathbb{E}_{\hat{y}_t \sim q_t} \right\rangle\!\!\right\rangle_{t=1}^{n} \left[ \sum_{t=1}^{n} \ell(\hat{y}_t, y_t) - \inf_{f \in \mathcal{F}} \left\{ \sum_{t=1}^{n} \ell(f(x_t), y_t) + \mathcal{B}_n(f; x_{1:n}, y_{1:n}) \right\} \right]
$$

$$
= \left\langle\!\!\left\langle \sup_{x_t \in \mathcal{X}} \sup_{p_t \in \Delta(\mathcal{Y})} \mathbb{E}_{y_t \sim p_t} \right\rangle\!\!\right\rangle_{t=1}^{n} \left[ \sup_{f \in \mathcal{F}} \left\{ \sum_{t=1}^{n} \inf_{\hat{y}_t \in \mathcal{D}} \mathbb{E}_{y_t \sim p_t} \left[ \ell(\hat{y}_t, y_t) \right] - \sum_{t=1}^{n} \ell(f(x_t), y_t) - \mathcal{B}_n(f; x_{1:n}, y_{1:n}) \right\} \right]
$$

$$
\le \left\langle\!\!\left\langle \sup_{x_t \in \mathcal{X}} \sup_{p_t \in \Delta(\mathcal{Y})} \mathbb{E}_{y_t \sim p_t} \right\rangle\!\!\right\rangle_{t=1}^{n} \left[ \sup_{f \in \mathcal{F}} \left\{ \sum_{t=1}^{n} \mathbb{E}_{y_t' \sim p_t} \left[ \ell(f(x_t), y_t') \right] - \ell(f(x_t), y_t) - \mathcal{B}_n(f; x_{1:n}, y_{1:n}) \right\} \right]. \quad (10)
$$

See [21] for more details of the steps involved in obtaining the above equality. Form this point on we split the proof for Equations 3 and 4. To prove the bound in Equation 3, note that, $\mathcal{B}_n(f; x_{1:n}, y_{1:n}) = \mathcal{B}_n(y_{1:n})$ and so, (this proof is similar in spirit to the one in [7])

$$
\mathcal{A}_n(\mathcal{F}) \le \left\langle\!\!\left\langle \sup_{x_t \in \mathcal{X}} \sup_{p_t \in \Delta(\mathcal{Y})} \mathbb{E}_{y_t \sim p_t} \right\rangle\!\!\right\rangle_{t=1}^{n} \left[ \sup_{f \in \mathcal{F}} \left\{ \sum_{t=1}^{n} \mathbb{E}_{y_t' \sim p_t} \left[ \ell(f(x_t), y_t') \right] - \ell(f(x_t), y_t) \right\} - \mathcal{B}_n(y_{1:n}) \right]
$$

$$
= \left\langle\!\!\left\langle \sup_{x_t \in \mathcal{X}} \sup_{p_t \in \Delta(\mathcal{Y})} \mathbb{E}_{y_t \sim p_t} \right\rangle\!\!\right\rangle_{t=1}^{n} \left[ \sup_{f \in \mathcal{F}} \left\{ \sum_{t=1}^{n} \mathbb{E}_{y_t' \sim p_t} \left[ \ell(f(x_t), y_t') \right] - \ell(f(x_t), y_t) \right\} \right.
$$
$$
\left. - \frac{1}{2} \mathcal{B}_n(y_{1:n}) - \frac{1}{2} \mathcal{B}_n(y_{1:n}) \right].
$$

Using linearity of expectation repeatedly (since $\mathcal{B}_n$ is independent of $f$ and $x_t$'s ),

$$
\mathcal{A}_n(\mathcal{F}) \le \left\langle\!\!\left\langle \sup_{x_t \in \mathcal{X}} \sup_{p_t \in \Delta(\mathcal{Y})} \mathbb{E}_{y_t \sim p_t} \right\rangle\!\!\right\rangle_{t=1}^{n} \left[ \sup_{f \in \mathcal{F}} \left\{ \sum_{t=1}^{n} \mathbb{E}_{y_t' \sim p_t} \left[ \ell(f(x_t), y_t') \right] - \ell(f(x_t), y_t) \right\} \right.
$$
$$
\left. - \frac{1}{2} \mathcal{B}_n(y_{1:n}) - \frac{1}{2} \mathbb{E}_{y_{1:n}' \sim p_{1:n}} \left[ \mathcal{B}_n(y_{1:n}') \right] \right].
$$

By Jensen's inequality, we pull out the expectations w.r.t. $y_t'$'s to further upper bound the above quantity by

$$\left\langle\!\!\left\langle \sup_{x_t \in \mathcal{X}} \sup_{p_t \in \Delta(\mathcal{Y})} \mathbb{E}_{y_t, y_t' \sim p_t} \right\rangle\!\!\right\rangle_{t=1}^{n} \left[ \sup_{f \in \mathcal{F}} \left\{ \sum_{t=1}^{n} \ell(f(x_t), y_t') - \ell(f(x_t), y_t) \right\} - \frac{1}{2}\mathcal{B}_n(y_{1:n}) - \frac{1}{2}\mathcal{B}_n(y_{1:n}') \right]$$

$$= \left\langle\!\!\left\langle \sup_{x_t \in \mathcal{X}} \sup_{p_t \in \Delta(\mathcal{Y})} \mathbb{E}_{y_t, y_t' \sim p_t} \mathbb{E}_{\epsilon_t} \right\rangle\!\!\right\rangle_{t=1}^{n} \left[ \sup_{f \in \mathcal{F}} \left\{ \sum_{t=1}^{n} \epsilon_t \left( \ell(f(x_t), y_t') - \ell(f(x_t), y_t) \right) \right\} - \frac{1}{2}\mathcal{B}_n(y_{1:n}) - \frac{1}{2}\mathcal{B}_n(y_{1:n}') \right]$$

$$\leq \left\langle\!\!\left\langle \sup_{x_t \in \mathcal{X}} \sup_{y_t, y_t' \in \mathcal{Y}} \mathbb{E}_{\epsilon_t} \right\rangle\!\!\right\rangle_{t=1}^{n} \left[ \sup_{f \in \mathcal{F}} \left\{ \sum_{t=1}^{n} \epsilon_t \left( \ell(f(x_t), y_t') - \ell(f(x_t), y_t) \right) \right\} - \frac{1}{2}\mathcal{B}_n(y_{1:n}) - \frac{1}{2}\mathcal{B}_n(y_{1:n}') \right]$$

$$\leq \left\langle\!\!\left\langle \sup_{x_t \in \mathcal{X}} \sup_{y_t \in \mathcal{Y}} \mathbb{E}_{\epsilon_t} \right\rangle\!\!\right\rangle_{t=1}^{n} \left[ \sup_{f \in \mathcal{F}} \left\{ \sum_{t=1}^{n} 2\epsilon_t \ell(f(x_t), y_t) \right\} - \mathcal{B}_n(y_{1:n}) \right]$$

$$= \sup_{\mathbf{x}, \mathbf{y}} \mathbb{E}_{\epsilon} \left[ \sup_{f \in \mathcal{F}} \left\{ 2 \sum_{t=1}^{n} \epsilon_t \ell(f(\mathbf{x}_t(\epsilon)), \mathbf{y}_t(\epsilon)) \right\} - \mathcal{B}_n(\mathbf{y}_{1:n}(\epsilon)) \right].$$

where the last but one step is by sub-additivity of supremum and linearity of expectation and last step is by skolemizing the supremum interleaved with average w.r.t. Rademacher random variables in the binary tree format.

We now move to proving Eq. (4). We start from Eq. (10):

$$\mathcal{A}_n(\mathcal{F}) \leq \left\langle\!\!\left\langle \sup_{x_t \in \mathcal{X}} \sup_{p_t \in \Delta(\mathcal{Y})} \mathbb{E}_{y_t \sim p_t} \right\rangle\!\!\right\rangle_{t=1}^{n} \left[ \sup_{f \in \mathcal{F}} \left\{ \sum_{t=1}^{n} \mathbb{E}_{y_t' \sim p_t} \left[ \ell(f(x_t), y_t') \right] - \ell(f(x_t), y_t) - \mathcal{B}_n(f; x_{1:n}, y_{1:n}) \right\} \right].$$

Using Jensen's inequality to pull out the expectations w.r.t. $y_t'$'s, we get

$$\leq \left\langle\!\!\left\langle \sup_{x_t \in \mathcal{X}} \sup_{p_t \in \Delta(\mathcal{Y})} \mathbb{E}_{y_t, y_t' \sim p_t} \right\rangle\!\!\right\rangle_{t=1}^{n} \left[ \sup_{f \in \mathcal{F}} \left\{ \sum_{t=1}^{n} \ell(f(x_t), y_t') - \ell(f(x_t), y_t) - \mathcal{B}_n(f; x_{1:n}, y_{1:n}) \right\} \right]$$

$$\leq \left\langle\!\!\left\langle \sup_{x_t \in \mathcal{X}} \sup_{p_t \in \Delta(\mathcal{Y})} \mathbb{E}_{y_t, y_t' \sim p_t} \sup_{y_t'' \in \mathcal{Y}} \right\rangle\!\!\right\rangle_{t=1}^{n} \left[ \sup_{f \in \mathcal{F}} \left\{ \sum_{t=1}^{n} \ell(f(x_t), y_t') - \ell(f(x_t), y_t) - \mathcal{B}_n(f; x_{1:n}, y_{1:n}'') \right\} \right]$$

$$= \left\langle\!\!\left\langle \sup_{x_t \in \mathcal{X}} \sup_{p_t \in \Delta(\mathcal{Y})} \mathbb{E}_{y_t, y_t' \sim p_t} \mathbb{E}_{\epsilon_t} \sup_{y_t'' \in \mathcal{Y}} \right\rangle\!\!\right\rangle_{t=1}^{n} \left[ \sup_{f \in \mathcal{F}} \left\{ \sum_{t=1}^{n} \epsilon_t \left( \ell(f(x_t), y_t') - \ell(f(x_t), y_t) \right) - \mathcal{B}_n(f; x_{1:n}, y_{1:n}'') \right\} \right]$$

$$\leq \left\langle\!\!\left\langle \sup_{x_t \in \mathcal{X}} \sup_{y_t, y_t' \in \mathcal{Y}} \mathbb{E}_{\epsilon_t} \sup_{y_t'' \in \mathcal{Y}} \right\rangle\!\!\right\rangle_{t=1}^{n} \left[ \sup_{f \in \mathcal{F}} \left\{ \sum_{t=1}^{n} \epsilon_t \left( \ell(f(x_t), y_t') - \ell(f(x_t), y_t) \right) - \mathcal{B}_n(f; x_{1:n}, y_{1:n}'') \right\} \right]$$

$$\leq \left\langle\!\!\left\langle \sup_{x_t \in \mathcal{X}} \sup_{y_t \in \mathcal{Y}} \mathbb{E}_{\epsilon_t} \sup_{y_t'' \in \mathcal{Y}} \right\rangle\!\!\right\rangle_{t=1}^{n} \left[ \sup_{f \in \mathcal{F}} \left\{ \sum_{t=1}^{n} 2\epsilon_t \ell(f(x_t), y_t) - \mathcal{B}_n(f; x_{1:n}, y_{1:n}'') \right\} \right]$$

$$= \sup_{\mathbf{x}, \mathbf{y}, \mathbf{y}'} \mathbb{E}_{\epsilon} \left[ \sup_{f \in \mathcal{F}} \left\{ 2 \sum_{t=1}^{n} \epsilon_t \ell(f(\mathbf{x}_t(\epsilon)), \mathbf{y}_t(\epsilon)) - \mathcal{B}_n(f; \mathbf{x}_{1:n}(\epsilon), \mathbf{y}_{2:n+1}'(\epsilon)) \right\} \right],$$

where in the last step we switch to tree notation, but keep in mind that each $y_t''$ is picked after drawing $\epsilon_t$, and thus the tree $\mathbf{y}'$ appears with one index shifted.

Finally, we proceed to prove inequality (5). Here, we employ the convexity assumption $\ell(\hat{y}_t, y_t) - \ell(f(x_t), y_t) \leq \ell'(\hat{y}_t, y_t)(\hat{y}_t - f(x_t))$, where the derivative is with respect to the first argument. As before, applying the minimax theorem,

$$\mathcal{A}_n(\mathcal{F}) = \left\langle\!\!\left\langle \sup_{x_t \in \mathcal{X}} \inf_{q_t \in \Delta(\mathcal{D})} \sup_{y_t \in \mathcal{Y}} \mathbb{E}_{\hat{y}_t \sim q_t} \right\rangle\!\!\right\rangle_{t=1}^{n} \left[ \sum_{t=1}^{n} \ell(\hat{y}_t, y_t) - \inf_{f \in \mathcal{F}} \left\{ \sum_{t=1}^{n} \ell(f(x_t), y_t) + \mathcal{B}_n(f; x_{1:n}, y_{1:n}) \right\} \right]$$

$$= \left\langle\!\!\left\langle \sup_{x_t \in \mathcal{X}} \sup_{p_t \in \Delta(\mathcal{Y})} \inf_{\hat{y}_t \in \mathcal{D}} \mathbb{E}_{y_t \sim p_t} \right\rangle\!\!\right\rangle_{t=1}^{n} \left[ \sum_{t=1}^{n} \ell(\hat{y}_t, y_t) - \inf_{f \in \mathcal{F}} \left\{ \sum_{t=1}^{n} \ell(f(x_t), y_t) + \mathcal{B}_n(f; x_{1:n}, y_{1:n}) \right\} \right]$$

$$\leq \left\langle\!\!\left\langle \sup_{x_t \in \mathcal{X}} \sup_{p_t \in \Delta(\mathcal{Y})} \inf_{\hat{y}_t \in \mathcal{D}} \mathbb{E}_{y_t \sim p_t} \right\rangle\!\!\right\rangle_{t=1}^{n} \left[ \sup_{f \in \mathcal{F}} \left\{ \sum_{t=1}^{n} \ell'(\hat{y}_t, y_t)(\hat{y}_t - f(x_t)) - \mathcal{B}_n(f; x_{1:n}, y_{1:n}) \right\} \right].$$

We may now pick $\hat{y}_t = \hat{y}_t^*(p_t) \triangleq \arg\min_{\hat{y}} \mathbb{E}_{y_t \sim p_t} \left[ \ell(\hat{y}_t, y_t) \right]$. By convexity (and assuming the loss allows swapping of derivative and expectation), $\mathbb{E}_{y_t \sim p_t} \left[ \ell'(\hat{y}_t, y_t) \right] = 0$. This (sub)optimal strategy yields an upper bound of

$$\left\langle\!\!\left\langle \sup_{x_t \in \mathcal{X}} \sup_{p_t \in \Delta(\mathcal{Y})} \mathbb{E}_{y_t \sim p_t} \right\rangle\!\!\right\rangle_{t=1}^{n} \left[ \sup_{f \in \mathcal{F}} \left\{ \sum_{t=1}^{n} \left( \ell'(\hat{y}_t^*, y_t) - \mathbb{E}_{y_t' \sim p_t} \left[ \ell'(\hat{y}_t^*, y_t') \right] \right)(\hat{y}_t^* - f(x_t)) - \mathcal{B}_n(f; x_{1:n}, y_{1:n}) \right\} \right].$$

Since $\left(\ell'(\hat{y}_t^*, y_t) - \mathbb{E}_{y_t' \sim p_t}\left[\ell'(\hat{y}_t^*, y_t')\right]\right)\hat{y}_t^*$ is independent of $f$ and has expected value of $0$, the above quantity is equal to

$$
\left\langle\!\!\left\langle \sup_{x_t \in \mathcal{X}} \sup_{p_t \in \Delta(\mathcal{Y})} \mathbb{E}_{y_t \sim p_t} \right\rangle\!\!\right\rangle_{t=1}^n \left[ \sup_{f \in \mathcal{F}} \left\{ \sum_{t=1}^n \left( \mathbb{E}_{y_t' \sim p_t}\left[\ell'(\hat{y}_t^*, y_t')\right] - \ell'(\hat{y}_t^*, y_t)\right) f(x_t) - \mathcal{B}_n(f; x_{1:n}, y_{1:n}) \right\} \right]
$$

$$
\leq \left\langle\!\!\left\langle \sup_{x_t \in \mathcal{X}} \sup_{p_t \in \Delta(\mathcal{Y})} \mathbb{E}_{y_t, y_t' \sim p_t} \right\rangle\!\!\right\rangle_{t=1}^n \left[ \sup_{f \in \mathcal{F}} \left\{ \sum_{t=1}^n \left(\ell'(\hat{y}_t^*, y_t') - \ell'(\hat{y}_t^*, y_t)\right) f(x_t) - \mathcal{B}_n(f; x_{1:n}, y_{1:n}) \right\} \right]
$$

$$
= \left\langle\!\!\left\langle \sup_{x_t \in \mathcal{X}} \sup_{p_t \in \Delta(\mathcal{Y})} \mathbb{E}_{y_t, y_t' \sim p_t} \mathbb{E}_{\epsilon_t} \right\rangle\!\!\right\rangle_{t=1}^n \left[ \sup_{f \in \mathcal{F}} \left\{ \sum_{t=1}^n \epsilon_t \left(\ell'(\hat{y}_t^*, y_t') - \ell'(\hat{y}_t^*, y_t)\right) f(x_t) - \mathcal{B}_n(f; x_{1:n}, y_{1:n}) \right\} \right].
$$

Replacing $\left(\ell'(\hat{y}_t^*, y_t') - \ell'(\hat{y}_t^*, y_t)\right)$ by $2Ls_t$ for $s_t \in [-1, 1]$ and taking supremum over $s_t$ we get,

$$
\leq \left\langle\!\!\left\langle \sup_{x_t \in \mathcal{X}} \sup_{p_t \in \Delta(\mathcal{Y})} \mathbb{E}_{y_t, y_t' \sim p_t} \sup_{s_t \in [-1,1]} \mathbb{E}_{\epsilon_t} \right\rangle\!\!\right\rangle_{t=1}^n \left[ \sup_{f \in \mathcal{F}} \left\{ \sum_{t=1}^n 2L\epsilon_t s_t f(x_t) - \mathcal{B}_n(f; x_{1:n}, y_{1:n}) \right\} \right]
$$

$$
\leq \left\langle\!\!\left\langle \sup_{x_t \in \mathcal{X}} \sup_{y_t} \sup_{s_t \in [-1,1]} \mathbb{E}_{\epsilon_t} \right\rangle\!\!\right\rangle_{t=1}^n \left[ \sup_{f \in \mathcal{F}} \left\{ \sum_{t=1}^n 2L\epsilon_t s_t f(x_t) - \mathcal{B}_n(f; x_{1:n}, y_{1:n}) \right\} \right].
$$

Since the suprema over $s_t$ are achieved at $\{\pm 1\}$ by convexity, the last expression is equal to

$$
\left\langle\!\!\left\langle \sup_{x_t \in \mathcal{X}} \sup_{y_t} \sup_{s_t \in \{-1,1\}} \mathbb{E}_{\epsilon_t} \right\rangle\!\!\right\rangle_{t=1}^n \left[ \sup_{f \in \mathcal{F}} \left\{ \sum_{t=1}^n 2L\epsilon_t s_t f(x_t) - \mathcal{B}_n(f; x_{1:n}, y_{1:n}) \right\} \right]
$$

$$
= \left\langle\!\!\left\langle \sup_{x_t \in \mathcal{X}} \sup_{y_t} \mathbb{E}_{\epsilon_t} \right\rangle\!\!\right\rangle_{t=1}^n \left[ \sup_{f \in \mathcal{F}} \left\{ \sum_{t=1}^n 2L\epsilon_t f(x_t) - \mathcal{B}_n(f; x_{1:n}, y_{1:n}) \right\} \right]
$$

$$
= \sup_{\mathbf{x}, \mathbf{y}} \mathbb{E}_\epsilon \left[ \sup_{f \in \mathcal{F}} \left\{ \sum_{t=1}^n 2L\epsilon_t f(\mathbf{x}_t(\epsilon)) - \mathcal{B}_n(f; \mathbf{x}_{1:n}(\epsilon), \mathbf{y}_{1:n}(\epsilon)) \right\} \right].
$$

In the last but one step we removed $s_t$, since for any function $\Psi$, and any $s \in \{\pm 1\}$, $\mathbb{E}\left[\Psi(s\epsilon)\right] = \frac{1}{2}\left(\Psi(s) + \Psi(-s)\right) = \frac{1}{2}\left(\Psi(1) + \Psi(-1)\right) = \mathbb{E}\left[\Psi(\epsilon)\right]$. $\qquad\square$

***Proof of Proposition 2.*** Define $Z_i = \left[X_i - B_i\theta_i\right]_+$. As long as $\theta_i \geq 1$, for any strictly positive $\tau$ we have the tail behavior

$$
P(Z_i \geq t) = P(X_i - B_i\theta_i \geq \tau) \leq C_1 \exp\left(-\frac{(B_i(\theta_i - 1) + \tau)^2}{2\sigma_i^2}\right) + C_2 \exp\left(-(B_i(\theta_i - 1) + \tau)s_i\right).
$$

Note that for any positive sequence $(\delta_i)_{i \in I}$ with $\delta = \sum_{i \in I} \delta_i$,

$$
\mathbb{E}\left[\sup_{i \in I}\{X_i - B_i\theta_i\}\right] \leq \mathbb{E}\left[\sup_{i \in I} Z_i\right] \leq \sum_{i \in I} \mathbb{E}\left[Z_i\right] \leq \delta + \sum_{i \in I} \int_{\delta_i}^\infty P(Z_i \geq \tau)d\tau.
$$

The sum of the integrals above is equal to

$$
\sum_{i \in I} \int_{\delta_i}^\infty P(X_i - B_i\theta_i \geq \tau)d\tau
$$

$$
\leq C_1 \sum_{i \in I} \int_0^\infty \exp\left(-\frac{(B_i(\theta_i - 1) + \tau)^2}{2\sigma_i^2}\right)dt + C_2 \sum_{i \in I} \int_0^\infty \exp\left(-(B_i(\theta_i - 1) + \tau)s_i\right)d\tau
$$

$$
\leq C_1 \sum_{i \in I} \exp\left(-\frac{1}{2}\left(\frac{B_i}{\sigma_i}\right)^2(\theta_i - 1)^2\right) \int_0^\infty e^{-\frac{\tau^2}{2\sigma_i^2}}d\tau + C_2 \sum_{i \in I} \exp\left(-B_is_i(\theta_i - 1)\right) \int_0^\infty e^{-\tau s_i}d\tau
$$

$$
\leq \sqrt{\frac{\pi}{2}}C_1 \sum_{i \in I} \sigma_i \exp\left(-\frac{1}{2}\left(\frac{B_i}{\sigma_i}\right)^2(\theta_i - 1)^2\right) + C_2 \sum_{i \in I} s_i^{-1} \exp\left(-B_is_i(\theta_i - 1)\right)
$$

$$
\leq \frac{\pi^2\sqrt{\pi}}{6\sqrt{2}}C_1\bar{\sigma} + \frac{\pi^2}{6}C_2(\bar{s})^{-1},
$$

where the last step is obtained by plugging in

$$
\theta_i = \max\left\{\frac{\sigma_i}{B_i}\sqrt{2\log(\sigma_i/\bar{\sigma}) + 4\log(i)}, (B_is_i)^{-1}\log\left(i^2(\bar{s}/s_i)\right)\right\} + 1
$$

and using as an upper bound $\frac{\sigma_i}{B_i}\sqrt{2\log(i^2\sigma_i/\bar{\sigma})} + 1$ for $\theta_i$ in the sub-gaussian part and $(B_is_i)^{-1}\log\left(i^2\bar{s}/s_i\right) + 1$ for $\theta_i$ in the sub-exponential part. Since $\delta$ can be chosen arbitrarily small, we may over-bound the above constant and obtain the result. $\qquad\square$

***Proof of Lemma 5.*** Fix $\gamma > 0$. For $j \geq 0$, let $V_j$ be a minimal sequential cover of $\mathcal{G}$ on $\mathbf{z}$ at scale $\beta_j = 2^{-j}\gamma$ and with respect to empirical $\ell_2$ norm. Let $\mathbf{v}^j[g,\epsilon]$ be an element guaranteed to be $\beta_j$-close to $f$ at the $j$-th level, for the given $\epsilon$. Choose $N = \log_2(2\gamma n)$, so that $\beta_N n \leq 1$. Let us use the shorthand $\mathcal{N}_2(\gamma) \triangleq \mathcal{N}_2(\mathcal{G}, \gamma, \mathbf{z})$.

For any $\epsilon \in \{\pm 1\}^n$ and $g \in \mathcal{G}$,

$$\sum_{t=1}^{n} \epsilon_t g(\mathbf{z}_t(\epsilon)) - 2\alpha g(\mathbf{z}_t(\epsilon))^2$$

can be written as

$$\sum_{t=1}^{n} \left(\epsilon_t(g(\mathbf{z}_t(\epsilon)) - \mathbf{v}_t^0[g,\epsilon](\epsilon))\right) + \sum_{t=1}^{n} \left(\epsilon_t \mathbf{v}_t^0[g,\epsilon](\epsilon) - 2\alpha g(\mathbf{z}_t(\epsilon))^2\right)$$

$$\leq \sum_{t=1}^{n} \left(\epsilon_t(g(\mathbf{z}_t(\epsilon)) - \mathbf{v}_t^0[g,\epsilon](\epsilon))\right) + \sum_{t=1}^{n} \left(\epsilon_t \mathbf{v}_t^0[g,\epsilon](\epsilon) - \alpha \mathbf{v}_t^0[g,\epsilon](\epsilon)^2\right)$$

$$= \sum_{t=1}^{n} \left(\epsilon_t(g(\mathbf{z}_t(\epsilon)) - \mathbf{v}_t^N[g,\epsilon](\epsilon))\right) + \sum_{t=1}^{n} \sum_{k=1}^{N} \epsilon_t \left(\mathbf{v}_t^k[g,\epsilon](\epsilon) - \mathbf{v}_t^{k-1}[g,\epsilon](\epsilon)\right)$$

$$+ \sum_{t=1}^{n} \left(\epsilon_t \mathbf{v}_t^0[g,\epsilon](\epsilon) - \alpha \mathbf{v}_t^0[g,\epsilon](\epsilon)^2\right).$$

By Cauchy-Schwartz, the first term is upper bounded by $n\beta_N \leq 1$. The second term above is upper bounded by

$$\sum_{k=1}^{N} \sum_{t=1}^{n} \epsilon_t \left(\mathbf{v}_t^k[g,\epsilon](\epsilon) - \mathbf{v}_t^{k-1}[g,\epsilon](\epsilon)\right) \leq \sum_{k=1}^{N} \sup_{\mathbf{w}^k \in W_k} \sum_{t=1}^{n} \epsilon_t \mathbf{w}_t^k(\epsilon),$$

where $W_k$ is a set of differences of trees for levels $k$ and $k-1$ (see [24, Proof of Theorem 3]). Finally, the third term is controlled by

$$\sum_{t=1}^{n} \left(\epsilon_t \mathbf{v}_t^0[g,\epsilon](\epsilon) - \alpha \mathbf{v}_t^0[g,\epsilon](\epsilon)^2\right) \leq \sup_{\mathbf{v} \in V_0} \sum_{t=1}^{n} \left(\epsilon_t \mathbf{v}_t(\epsilon) - \alpha \mathbf{v}_t^2(\epsilon)\right).$$

The probability in the statement of the Lemma can now be upper bounded by

$$P\left(\sum_{k=1}^{N} \sup_{\mathbf{w}^k \in W_k} \sum_{t=1}^{n} \epsilon_t \mathbf{w}_t^k(\epsilon) + \sup_{\mathbf{v} \in V_0} \sum_{t=1}^{n} \left(\epsilon_t \mathbf{v}_t(\epsilon) - \alpha \mathbf{v}_t^2(\epsilon)\right) - \frac{\log \mathcal{N}_2(\gamma)}{\alpha} - 12\sqrt{2} \int_{1/n}^{\gamma} \sqrt{n \log \mathcal{N}_2(\delta)} d\delta > \tau\right).$$

In view of

$$\sqrt{72} \sum_{k=1}^{N} \beta_k \sqrt{n \log \mathcal{N}_2(\beta_k)} \leq 12\sqrt{2} \int_{1/n}^{\gamma} \sqrt{n \log \mathcal{N}_2(\delta)} d\delta$$

this probability can be further upper bounded by

$$P\left(\sum_{k=1}^{N} \sup_{\mathbf{w}^k \in W_k} \sum_{t=1}^{n} \epsilon_t \mathbf{w}_t^k(\epsilon) + \sup_{\mathbf{v} \in V_0} \sum_{t=1}^{n} \left(\epsilon_t \mathbf{v}_t(\epsilon) - \alpha \mathbf{v}_t^2(\epsilon)\right) - \frac{\log \mathcal{N}_2(\gamma)}{\alpha} - \sqrt{72} \sum_{k=1}^{N} \beta_k \sqrt{n \log \mathcal{N}_2(\beta_k)} > \tau\right).$$

Define a distribution $p$ on $\{1, \ldots, N\}$ by $p_k = \frac{\beta_k \sqrt{n \log \mathcal{N}_2(\beta_k)}}{\sum_{k=1}^{N} \beta_j \sqrt{n \log \mathcal{N}_2(\beta_j)}}$. Then the above probability can be upper bounded by

$$P\left(\exists k \in [N] \text{ s.t. } \sup_{\mathbf{w}^k \in W_k} \sum_{t=1}^{n} \epsilon_t \mathbf{w}_t^k(\epsilon) - \sqrt{72}\beta_k \sqrt{n \log \mathcal{N}_2(\beta_k)} > \frac{\tau p_k}{2}\right.$$

$$\left. \vee \sup_{\mathbf{v} \in V_0} \sum_{t=1}^{n} \left(\epsilon_t \mathbf{v}_t(\epsilon) - \alpha \mathbf{v}_t^2(\epsilon)\right) - \frac{\log \mathcal{N}_2(\gamma)}{\alpha} > \frac{\tau}{2}\right)$$

$$\leq \sum_{k=1}^{N} P\left(\sup_{\mathbf{w}^k \in W_k} \sum_{t=1}^{n} \epsilon_t \mathbf{w}_t^k(\epsilon) - \sqrt{72}\beta_k \sqrt{n \log \mathcal{N}_2(\beta_k)} > \frac{\tau p_k}{2}\right)$$

$$+ P\left(\sup_{\mathbf{v} \in V_0} \sum_{t=1}^{n} \left(\epsilon_t \mathbf{v}_t(\epsilon) - \alpha \mathbf{v}_t^2(\epsilon)\right) - \frac{\log \mathcal{N}_2(\gamma)}{\alpha} > \frac{\tau}{2}\right).$$

The second term can be upper bounded using Chernoff method by

$$\sum_{\mathbf{v} \in V_0} P\left(\sum_{t=1}^{n} \left(\epsilon_t \mathbf{v}_t(\epsilon) - \alpha \mathbf{v}_t^2(\epsilon)\right) - \frac{\log \mathcal{N}_2(\gamma)}{\alpha} > \frac{\tau}{2}\right)$$

$$\leq \mathcal{N}_2(\gamma) \exp\left(-\frac{\alpha\tau}{2} - \log \mathcal{N}_2(\gamma)\right) \leq \exp\left(-\frac{\alpha\tau}{2}\right)$$

while the first sum of probabilities can be upper bounded by

$$\sum_{k=1}^{N} \sum_{\mathbf{w}^k \in W_k} P\left( \sum_{t=1}^{n} \epsilon_t \mathbf{w}_t^k(\epsilon) - \sqrt{72}\beta_k \sqrt{n \log \mathcal{N}_2(\beta_k)} > \frac{\tau \beta_k \sqrt{n \log \mathcal{N}_2(\beta_k)}}{2 \sum_{k=1}^{N} \beta_k \sqrt{n \log \mathcal{N}_2(\beta_k)}} \right). \tag{11}$$

For any $k$, the tail probability above is controlled by Hoeffding-Azuma inequality as

$$P\left( \sum_{t=1}^{n} \epsilon_t \mathbf{w}_t^k(\epsilon) > \beta_k \sqrt{n \log \mathcal{N}_2(\beta_k)} \left( 6\sqrt{2} + \frac{\tau}{2 \sum_{k=1}^{N} \beta_k \sqrt{n \log \mathcal{N}_2(\beta_k)}} \right)^2 \right)$$

$$\leq \exp\left( -\frac{1}{18} \log \mathcal{N}_2(\beta_k) \left( 6\sqrt{2} + \frac{\tau}{2 \sum_{k=1}^{N} \beta_k \sqrt{n \log \mathcal{N}_2(\beta_k)}} \right)^2 \right)$$

$$\leq \exp\left( -4 \log \mathcal{N}_2(\beta_k) \right) \exp\left( -\frac{\tau^2}{18 \left( 2 \sum_{k=1}^{N} \beta_k \sqrt{n \log \mathcal{N}_2(\beta_k)} \right)^2} \right),$$

because $\frac{1}{n} \sum_{t=1}^{n} \mathbf{w}_t^k(\epsilon)^2 \leq 3\beta_k^2$ for any $\epsilon$ by triangle inequality (see [24]). Then the double sum in (11) is upper bounded by

$$\Gamma \exp\left( -\frac{\tau^2}{18 \left( 2 \sum_{k=1}^{N} \beta_k \sqrt{n \log \mathcal{N}_2(\beta_k)} \right)^2} \right),$$

where $\Gamma \geq \sum_{k=1}^{N} \mathcal{N}_2(\beta_k)^{-2}$. This upper bound can be further relaxed to

$$\Gamma \exp\left( -\frac{\tau^2}{2 \left( 12 \int_{1/n}^{\gamma} \sqrt{n \log \mathcal{N}_2(\delta)} d\delta \right)^2} \right).$$

Since $N = \log_2(2\gamma n)$, we may take

$$\Gamma = \sum_{k=1}^{\log_2(2\gamma n)} \mathcal{N}_2(\gamma 2^{-k})^{-2}.$$

$\square$

***Proof of Corollary 6.*** Let $\mathcal{N}_2(\gamma) \triangleq \mathcal{N}_2(\mathcal{G}, \gamma, \mathbf{z})$. Observe that

$$2\sqrt{2(\log n)(\log \mathcal{N}_2(\gamma/2))\left( \sum_{t=1}^{n} g^2(\mathbf{z}_t(\epsilon)) + 1 \right)} = \inf_{\alpha} \left\{ \frac{(\log n)(\log \mathcal{N}_2(\gamma/2))}{\alpha} + 2\alpha \left( \sum_{t=1}^{n} g^2(\mathbf{z}_t(\epsilon)) + 1 \right) \right\}$$

and, furthermore, the optimal $\alpha$ is

$$\sqrt{\frac{(\log n)(\log \mathcal{N}_2(\gamma/2))}{2(\sum_{t=1}^{n} g^2(\mathbf{z}_t(\epsilon)) + 1)}}$$

which is a number between $d_\ell = \sqrt{\frac{(\log n)(\log \mathcal{N}_2(\gamma/2))}{2(n+1)}}$ and $d_u = \sqrt{(\log n)(\log \mathcal{N}_2(\gamma/2))}$ as long as $\mathcal{N}_2(\gamma/2) > 1$. With this we get

$$\sup_{\substack{g \in \mathcal{G} \\ \gamma \in [n^{-1}, 1]}} \left[ \sum_{t=1}^{n} \epsilon_t g(\mathbf{z}_t(\epsilon)) - 4\sqrt{2(\log n)(\log \mathcal{N}_2(\gamma/2))\left( \sum_{t=1}^{n} g^2(\mathbf{z}_t(\epsilon)) + 1 \right)} \right.$$

$$\left. -24\sqrt{2} \log n \int_{1/n}^{\gamma} \sqrt{n \log \mathcal{N}_2(\delta)} d\delta + 2 \log n \right]$$

$$\leq \sup_{\substack{g \in \mathcal{G} \\ \gamma \in [n^{-1}, 1], \alpha \in [d_\ell, d_u]}} \left[ \sum_{t=1}^{n} \epsilon_t g(\mathbf{z}_t(\epsilon)) - \frac{2(\log n)(\log \mathcal{N}_2(\gamma/2))}{\alpha} - 4\alpha \sum_{t=1}^{n} g^2(\mathbf{z}_t(\epsilon)) \right.$$

$$\left. -24\sqrt{2} \log n \int_{1/n}^{\gamma} \sqrt{n \log \mathcal{N}_2(\delta)} d\delta - 2 \log n \right]. \tag{12}$$

The case of $\gamma \in [1/n, 2/n)$ will be considered separately. Let us assume $\gamma \geq 2/n$. We now discretize both $\alpha$ and $\gamma$ by defining $\alpha_i = 2^{-(i-1)}d_u$ and $\gamma_j = 2^j n^{-1}$, $i, j \geq 1$. We go to an upper bound by mapping each $\alpha$ to $\alpha_i$ or $\alpha_i/2$, depending on the direction of the sign. Similarly, we map $\gamma$ to either $\gamma_i$ or $2\gamma_i$. The upper bound becomes

$$\max_{i,j} \sup_{g \in \mathcal{G}} \sum_{t=1}^{n} \left(\epsilon_t g(\mathbf{z}_t(\epsilon)) - 2\alpha_i g^2(\mathbf{z}_t(\epsilon))\right) - (2\log n)\left(\frac{\log \mathcal{N}_2(\gamma_j)}{\alpha_i} + 12\sqrt{2}\int_{1/n}^{\gamma_j}\sqrt{n\log\mathcal{N}_2(\delta)}d\delta + 1\right).$$

Given the doubling nature of $\alpha_i$ and $\gamma_j$, the indices $i, j$ are upper bounded by $O(\log n)$. Now define a collection of random variables indexed by $(i, j)$

$$X_{i,j} = \sup_{g \in \mathcal{G}} \sum_{t=1}^{n} \epsilon_t g(\mathbf{z}_t(\epsilon)) - 2\alpha_i g^2(\mathbf{z}_t(\epsilon))$$

and constants

$$B_{i,j} = \frac{\log \mathcal{N}_2(\gamma_j)}{\alpha_i} + 12\sqrt{2}\int_{1/n}^{\gamma_j}\sqrt{n\log\mathcal{N}_2(\delta)}d\delta + 1.$$

Lemma 5 establishes that

$$P(X_{i,j} - B_{i,j} > \tau) \leq \Gamma \exp\left(-\frac{\tau^2}{2\sigma_j^2}\right) + \exp\left(-\frac{\alpha_i\tau}{2}\right)$$

where $\sigma_j = 12\sqrt{2}\int_{\frac{1}{n}}^{\gamma_j}\sqrt{n\log\mathcal{N}_2(\delta)}d\delta$ and $\Gamma$ as specified in Lemma 5. Whenever $\delta$-entropy grows as $\delta^{-p}$, $\sigma_j \leq 12\sqrt{2}\sqrt{n}$, ensuring $\log(\sigma_j/\sigma_1) \leq \log(n)$. Further, we can take $1 \leq \Gamma \leq \log(2n)$.

Proposition 2 is used with a sequence of random variables, but we can easily put the pairs $(i, j)$ into a vector of size at most $\log_2(n)^2$. Observe that $s_i = \alpha_i/2$, $(B_{i,j}s_i)^{-1} \leq 2$, $\sigma_j/B_{i,j} \leq 1$, $s_1/s_i \leq \sqrt{2(n+1)}$. Then, by taking $\bar{\sigma} = \min\{1/\Gamma, \sigma_1\}$ and $\bar{s} = s_1$,

$$\theta_{k_{i,j}} = \max\left\{\frac{\sigma_j}{B_{i,j}}\sqrt{2\log(\sigma_j/\bar{\sigma}) + 4\log(k_{i,j})}, (B_{i,j}s_i)^{-1}\log\left(k_{i,j}^2(\bar{s}/s_i)\right)\right\} + 1$$

$$\leq \max\left\{\sqrt{2\log(n) + 2\log(\log(2n)) + 4\log(k_{i,j})}, 2\log\left(k_{i,j}^2\sqrt{2(n+1)}\right)\right\} + 1$$

where $k_{i,j} = (\log n)\cdot(i-1) + j$. This choice of the multiplier ensures

$$\mathbb{E}\max_{i,j}\left\{X_{i,j} - \theta_{k_{i,j}}B_{i,j}\right\} \leq 3\Gamma\bar{\sigma} + 4\alpha_1^{-1} \leq 7$$

and $\theta_{i,j}$ is shown to be upper bounded by $2\log n$. Hence

$$\mathbb{E}\left[\sup_{g \in \mathcal{G}, \gamma}\sum_{t=1}^{n}\epsilon_t g(\mathbf{z}_t(\epsilon)) - 4\sqrt{2\log n\log\mathcal{N}_2(\gamma/2)\left(\sum_{t=1}^{n}g^2(\mathbf{z}_t(\epsilon)) + 1\right)} - 24\sqrt{2}\log n\int_{1/n}^{\gamma}\sqrt{n\log\mathcal{N}_2(\delta)}d\delta\right]$$

$$\leq 7 + 2\log n.$$

Now, consider the case $\gamma \in [1/n, 2/n)$. We upper bound (12) by

$$\max_i \sup_{g \in \mathcal{G}}\sum_{t=1}^{n}\left(\epsilon_t g(\mathbf{z}_t(\epsilon)) - 2\alpha_i g^2(\mathbf{z}_t(\epsilon))\right) - (2\log n)\left(\frac{\log\mathcal{N}_2(1/n)}{\alpha_i} + 1\right),$$

which is controlled by setting $\gamma = 1/n$ in Lemma 5. This case is completed by invoking Proposition 2 as before. $\square$

***Proof of Corollary 7.*** Assume $N > e$ and let $C > 0$. We first note that

$$\inf_{\alpha > 0}\left\{\frac{C\log\left(\frac{\sqrt{C}\log N}{\alpha}\right)\log N}{\alpha} + \alpha\left(\sum_{t=1}^{n}g^2(\mathbf{z}_t(\epsilon)) + \frac{e}{\log N}\right)\right\}$$

$$\leq 2\log\left(\log N\sum_{t=1}^{n}g^2(\mathbf{z}(\epsilon)) + e\right)\sqrt{C\left(\log N\sum_{t=1}^{n}g^2(\mathbf{z}(\epsilon)) + e\right)}$$

with the inequality obtained using $\alpha^\star = \sqrt{\frac{C\log N}{\Sigma_{t=1}^{n}g^2(\mathbf{z}(\epsilon)) + e/\log N}}$, which is a number between $d_\ell \triangleq \sqrt{\frac{C\log N}{n + e/\log N}}$ and $d_u \triangleq \sqrt{\frac{C}{e}}\log N$. Subsequently,

$$\sup_{g \in \mathcal{G}}\sum_{t=1}^{n}\epsilon_t g(\mathbf{z}_t(\epsilon)) - 2\log\left(\log N\sum_{t=1}^{n}g^2(\mathbf{z}(\epsilon)) + e\right)\sqrt{C\left(\log N\sum_{t=1}^{n}g^2(\mathbf{z}(\epsilon)) + e\right)}$$

$$\leq \sup_{\substack{g \in \mathcal{G} \\ \alpha \in [d_\ell, d_u]}}\left[\sum_{t=1}^{n}\epsilon_t g(\mathbf{z}_t(\epsilon)) - \alpha\sum_{t=1}^{n}g^2(\mathbf{z}_t(\epsilon)) - \frac{C\log N}{\alpha}\log\left(\frac{\sqrt{C}\log N}{\alpha}\right)\right].$$

Let $L = \left\lceil \log_2\left(\sqrt{\frac{n \log N}{e} + 1}\right) + 1 \right\rceil$. We discretize the range of $\alpha$ by defining $\alpha_i = d_u 2^{-(i-1)}$ for $i \in [L]$. The following upper bound holds:

$$\sup_{\substack{g \in \mathcal{G} \\ i \in [L]}} \left[ \sum_{t=1}^{n} \epsilon_t g(\mathbf{z}_t(\epsilon)) - \frac{\alpha_i}{2} \sum_{t=1}^{n} g^2(\mathbf{z}_t(\epsilon)) - \frac{C \log N}{\alpha_i} \log\left(\frac{\sqrt{C} \log N}{\alpha_i}\right) \right].$$

Define a collection of random variables indexed by $i \in [L]$ with

$$X_i = \sup_{g \in \mathcal{G}} \left[ \sum_{t=1}^{n} \epsilon_t g(\mathbf{z}_t(\epsilon)) - \frac{\alpha_i}{2} \sum_{t=1}^{n} g^2(\mathbf{z}_t(\epsilon)) \right]$$

and let $B_i = \frac{4 \log N}{\alpha_i}$. Applying Lemma 5 with $\gamma = 1/n$ establishes

$$P(X_i - B_i > \tau) \leq \exp\left(-\frac{\alpha_i \tau}{8}\right).$$

We now set $s_i = \alpha_i/8$ and $\bar{s} = s_1$, and apply Proposition 2, yielding

$$\mathbb{E}\{X_i - B_i \theta_i\} \leq \frac{16\sqrt{e}}{C}.$$

It remains to relate this quantity to the rate we are trying to achieve. Note that our bound on $P(X_i - B_i > \tau)$ has a pure exponential tail, so we only need to consider $\theta_i = (B_i s_i)^{-1} \log(i^2(\bar{s}/s_i)) + 1$. Taking $C \geq 32$ and observing that $(B_i s_i)^{-1} \leq 2$, we obtain

$$\theta_i = (B_i s_i)^{-1} \log(i^2(\bar{s}/s_i)) + 1 \leq 2 \log(i^2(\bar{s}/s_i)) + 1 = 2 \log(i^2 2^{i-1}) + 1 \leq 2 \log(i^2 2^i)$$

$$\leq \frac{C}{4} \log\left(\frac{\sqrt{C} \log N}{\alpha_i}\right).$$

Finally, we have

$$\sup_{\substack{g \in \mathcal{G} \\ i \in [L]}} \left[ \sum_{t=1}^{n} \epsilon_t g(\mathbf{z}_t(\epsilon)) - \frac{\alpha_i}{2} \sum_{t=1}^{n} g^2(\mathbf{z}_t(\epsilon)) - \frac{32 \log N}{\alpha_i} \log\left(\frac{\sqrt{32} \log N}{\alpha_i}\right) \right] \leq \mathbb{E}\{X_i - B_i \theta_i\} \leq \frac{\sqrt{e}}{2} \leq 1.$$

$\square$

***Proof of Corollary 8.*** We prove the corollary for convex Lipschitz loss where we remove the loss function using the symmetrization lemma shown earlier. However even if we consider non-convex classes, the loss is readily removed in the step in the proof below where we apply Lemma 4 where the Lipchitz constant is removed when we move to covering numbers. However this is a well known technique and to make the proof simpler we simply assume convexity of loss as well. Our starting point to proving the bounds is Lemma 1, Eq. (4). To show achievability it suffices to show that

$$\mathbb{E}_\epsilon \sup_{f \in \mathcal{F}} \sum_{t=1}^{n} \epsilon_t f(\mathbf{x}_t(\epsilon)) - K_1 \mathcal{R}_n(\mathcal{F}(2R(f))) \log^{3/2} n \left(1 + \sqrt{\log\left(\frac{\mathcal{R}_n(\mathcal{F}(2R(f)))}{\mathcal{R}_n(\mathcal{F}(R(1)))}\right) + \log(\log(2R(f)))}\right)$$

$$\leq K_2 \Gamma \mathcal{R}_n(\mathcal{F}(1)) \log^{3/2} n$$

where $\Gamma$ is the constant that will be inherited from Lemma 4. Define $R_i = 2^i$ and note that since the Rademacher complexity of the class $\mathcal{F}(R)$ is non-decreasing with $R$,

$$\sup_{f \in \mathcal{F}} \sum_{t=1}^{n} \epsilon_t f(\mathbf{x}_t(\epsilon)) - K_1 \mathcal{R}_n(\mathcal{F}(2R(f))) \log^{3/2} n \left(1 + \sqrt{\log\left(\frac{\mathcal{R}_n(\mathcal{F}(2R(f)))}{\mathcal{R}_n(\mathcal{F}(1))}\right) + \log(\log(2R(f)))}\right)$$

$$= \sup_{R \geq 1} \sup_{f \in \mathcal{F}(R)} \sum_{t=1}^{n} \epsilon_t f(\mathbf{x}_t(\epsilon)) - K_1 \mathcal{R}_n(\mathcal{F}(2R)) \log^{3/2} n \left(1 + \sqrt{\log\left(\frac{\mathcal{R}_n(\mathcal{F}(2R))}{\mathcal{R}_n(\mathcal{F}(1))}\right) + \log(\log(2R))}\right)$$

$$\leq \max_{i \in \mathbb{N}} \sup_{f \in \mathcal{F}(R_i)} \sum_{t=1}^{n} \epsilon_t f(\mathbf{x}_t(\epsilon)) - K_1 \mathcal{R}_n(\mathcal{F}(R_i)) \log^{3/2} n \left(1 + \sqrt{\log\left(\frac{\mathcal{R}_n(\mathcal{F}(R_i))}{\mathcal{R}_n(\mathcal{F}(1))}\right) + \log(\log(R_i))}\right).$$

(13)

Denote a shorthand $C_n = \sqrt{96 \log^3(en^2)}$ and $D_n^i = \mathcal{R}_n(\mathcal{F}(R_i))$. Now note that by Lemma 4 we have that for every $i$ and every $\theta > 1$,

$$P_\epsilon\left(\sup_{f \in \mathcal{F}(R_i)}\left|\sum_{t=1}^n \epsilon_t f(\mathbf{x}_t(\epsilon))\right| > 8\left(1 + \theta C_n\right) \cdot D_n^i\right) \le 2\Gamma e^{-3\theta^2}.$$

Let $X_i = \sup_{f \in \mathcal{F}(R_i)} |\sum_{t=1}^n \epsilon_t f(\mathbf{x}_t(\epsilon))|$ and let $B_i = 8\left(1 + C_n\right) \cdot D_n^i$. In this case rewriting the above one sided tail bound appropriately (with $\theta = 1 + \tau/(8 C_n D_n^i)$) we see that for any $\tau > 0$,

$$P(X_i - B_i > \tau) \le \frac{2\Gamma}{e^3} \exp\left(-\frac{\tau^2}{2^8 \log^3(en^2)\mathcal{R}_n^2(\mathcal{F}(R_i))}\right).$$

This establishes one-sided subgaussian tail behavior. Now applying Proposition 2 and setting $\theta_i$ as suggested by the proposition we conclude that

$$\mathbb{E}_\epsilon\left[\max_{i \in \mathbb{N}} \sup_{f \in \mathcal{F}(R_i)} \sum_{t=1}^n \epsilon_t f(\mathbf{x}_t(\epsilon)) - K_1 \mathcal{R}_n(\mathcal{F}(R_i)) \log^{3/2} n\left(1 + \sqrt{\log\left(\frac{\mathcal{R}_n(\mathcal{F}(R_i))}{\mathcal{R}_n(\mathcal{F}(1))}\right) + \log(\log(R_i))}\right)\right]$$

$$\le K_2 \Gamma \mathcal{R}_n(\mathcal{F}(1)) \log^{3/2} n.$$

This concludes the proof by appealing to Eq. (13). $\qquad\square$

### *Proof of Achievability for Example 4.2.*

**Lemma 10.** *The following bound is achievable in the setting of Example 4.2:*

$$\mathcal{B}(f) = D\sqrt{n}\left(8\|f\|\left(1 + \sqrt{\log(2\|f\|) + \log\log(2\|f\|)}\right) + 12\right).$$

This proof specializes the proof of Corollary 8 to the regime where Lemma 3 applies.

Recall our parameterization of $\mathcal{F}$: $\mathcal{F}(R) = \{f \in \mathcal{F} : \|f\| \le R\}$. It was shown in [26] that $\mathcal{C}_n(\mathcal{F}(R)) \doteq 2RD\sqrt{n}$ is an upper bound for $\mathcal{R}_n(\mathcal{F}(R))$. We consider the rate

$$\mathcal{B}_n(f) = 2\mathcal{C}_n(\mathcal{F}(2R(f)))\left(1 + \sqrt{\log\left(\frac{\mathcal{C}_n(\mathcal{F}(2R(f)))}{\mathcal{C}_n(\mathcal{F}(1))}\right) + \log\log_2(2R(f))}\right).$$

We begin by applying Lemma 1 (5), yielding

$$\mathcal{A}_n \le \sup_{\mathbf{y}} \mathbb{E}_\epsilon \sup_f 2\sum_{t=1}^n \epsilon_t \langle f, \mathbf{y}_t(\epsilon)\rangle - 2\mathcal{C}_n(\mathcal{F}(2R(f)))\left(1 + \sqrt{\log\left(\frac{\mathcal{C}_n(\mathcal{F}(2R(f)))}{\mathcal{C}_n(\mathcal{F}(1))}\right) + \log\log_2(2R(f))}\right).$$

We now discretize the range of $R$ via $R_i = 2^i$. By analogy with the proof of Corollary 8 we get the upper bound,

$$\sup_{\mathbf{y}} \mathbb{E}_\epsilon \sup_{i \in \mathbb{N}}\left[\sup_{f \in \mathcal{F}(R_i)} 2\sum_{t=1}^n \epsilon_t \langle f, \mathbf{y}_t(\epsilon)\rangle - 2\mathcal{C}_n(\mathcal{F}(R_i))\left(1 + \sqrt{\log\left(\frac{\mathcal{C}_n(\mathcal{F}(R_i))}{\mathcal{C}_n(\mathcal{F}(1))}\right) + \log\log_2(R_i)}\right)\right]$$

$$= \sup_{\mathbf{y}} \mathbb{E}_\epsilon \sup_{i \in \mathbb{N}}\left[2R_i\left\|\sum_{t=1}^n \epsilon_t \mathbf{y}_t(\epsilon)\right\|_\star - 4D\sqrt{n}R_i\sqrt{\log(R_i) + \log(i)}\right].$$

Fix a $\mathcal{Y}$-valued tree $\mathbf{y}$ and define a set of random variables $X_i = 2R_i\|\sum_{t=1}^n \epsilon_t \mathbf{y}_t(\epsilon)\|_\star$. Let $B_i = 2D\sqrt{n}R_i$. Lemma 3 shows that

$$P(X_i - B_i \ge \tau) \le 2\exp\left(-\frac{\tau^2}{8D^2 R_i^2 n}\right).$$

So we have $\sigma_i = 2DR_i\sqrt{n}$, and it will be sufficient to set $\bar{\sigma} = 2D\sqrt{n}$. Since our tail bound is purely sub-gaussian, we apply Proposition 2 with $\theta_i = \frac{\sigma_i}{B_i}\sqrt{2\log(\sigma_i/\bar{\sigma}) + 4\log(i)} + 1$, yielding the following bound:

$$\sup_{\mathbf{y}} \mathbb{E}_\epsilon \sup_{i \in \mathbb{N}}\left[2R_i\left\|\sum_{t=1}^n \epsilon_t \mathbf{y}_t(\epsilon)\right\|_\star - 4D\sqrt{n}R_i\sqrt{\log(R_i) + \log(i)}\right] \le 12D\sqrt{n}.$$

$$\square$$

### *Proof of Achievability for Example 4.5.* Unfortunately, the general symmetrization proof in Lemma 1 does not suffice for this problem. In what follows we use a more specialized symmetrization technique to prove the lemma.

**Lemma 11.** *For any countable class of experts, when we consider $\mathcal{F}$ to be the class of all distributions over the set of experts, the following adaptive bound is achievable:*

$$\mathcal{B}_n(f; y_{1:n}) = \sqrt{50 \left(\mathrm{KL}(f|\pi) + \log(n)\right) \sum_{t=1}^{n} \langle f, y_t \rangle + 50 \left(\mathrm{KL}(f|\pi) + \log(n)\right) + 1}.$$

To show that the rate is achievable we need to show that $\mathcal{A}_n \leq 0$. Since each $\hat{y}_t$ is a distribution over experts and we are in the linear setting, we do not need to randomize in the definition of the minimax value. Let us use the shorthand

$$C(f) = \mathrm{KL}(f|\pi) + \log(n),$$

and take constants $K_1$, $K_2$ to be determined later. Define

$$\mathcal{A}_n = \left\langle\!\!\!\left\langle \inf_{\hat{y}_t \in \Delta} \sup_{y_t \in \mathcal{Y}} \right\rangle\!\!\!\right\rangle_{t=1}^{n} \left[ \sum_{t=1}^{n} \langle \hat{y}_t, y_t \rangle - \inf_{f \in \Delta} \left\{ \sum_{t=1}^{n} \langle f, y_t \rangle + \sqrt{KC(f) \sum_{t=1}^{n} \mathbb{E}_{i \sim f} \langle e_i, y_t \rangle^2} + \sqrt{K'}C(f) \right\} \right].$$

Using repeated minimax swap, this expression is equal to

$$\left\langle\!\!\!\left\langle \sup_{p_t \in \Delta(\mathcal{Y})} \inf_{\hat{y}_t \in \Delta} \right\rangle\!\!\!\right\rangle_{t=1}^{n} \left[ \sum_{t=1}^{n} \langle \hat{y}_t, y_t \rangle - \inf_{f \in \Delta} \left\{ \sum_{t=1}^{n} \langle f, y_t \rangle + \sqrt{KC(f) \sum_{t=1}^{n} \mathbb{E}_{i \sim f} \langle e_i, y_t \rangle^2} + \sqrt{K'}C(f) \right\} \right]$$

$$= \left\langle\!\!\!\left\langle \sup_{p_t \in \Delta(\mathcal{Y})} \mathbb{E}_{y_t \sim p_t} \right\rangle\!\!\!\right\rangle_{t=1}^{n} \left[ \sum_{t=1}^{n} \inf_{\hat{y}_t \in \Delta} \mathbb{E}_{y_t \sim p_t} \left[ \langle \hat{y}_t, y_t \rangle \right] \right.$$

$$\left. - \inf_{f \in \Delta} \left\{ \sum_{t=1}^{n} \langle f, y_t \rangle + \sqrt{KC(f) \sum_{t=1}^{n} \mathbb{E}_{i \sim f} \langle e_i, y_t \rangle^2} + \sqrt{K'}C(f) \right\} \right].$$

By sub-additivity of square-root we pass to an upper bound,

$$\left\langle\!\!\!\left\langle \sup_{p_t} \mathbb{E}_{y_t \sim p_t} \right\rangle\!\!\!\right\rangle_{t=1}^{n} \left[ \sup_{f \in \mathcal{F}} \sum_{t=1}^{n} \inf_{\hat{y}_t \in \Delta} \mathbb{E}_{y_t \sim p_t} \left[ \langle \hat{y}_t, y_t \rangle \right] - \mathbb{E}_{e_i \sim f} \left[ \langle e_i, y_t \rangle \right] \right.$$

$$\left. - \sqrt{C(f) \left( K \sum_{t=1}^{n} \mathbb{E}_{i \sim f} \left[ \langle e_i, y_t \rangle^2 \right] + K'C(f) \right)} \right].$$

We now split the square root according to the formula $\sqrt{ab} = \inf_{\alpha > 0} \{a/2\alpha + \alpha b/2\}$ and note the range of the optimal value:

$$\frac{1}{\sqrt{n}} \leq \alpha^* = \sqrt{\frac{C(f)}{\left( K \sum_{t=1}^{n} \mathbb{E}_{i \sim f} \left[ \langle e_i, y_t \rangle^2 \right] + K'C(f) \right)}} \leq \frac{1}{\sqrt{K'}}. \tag{14}$$

Let us discretize the interval by setting $\alpha_i = \frac{1}{\sqrt{K'}} 2^{-(i-1)}$ for $i = 1, \ldots, N$ and note that we only need to take $N = O(\log(n))$ elements. Write $I = \{\alpha_1, \ldots, \alpha_N\}$. Observe that

$$\sqrt{ab} = \inf_{\alpha > 0} \{a/2\alpha + \alpha b/2\} \geq \min_{\alpha \in I} \{a/4\alpha + \alpha b/2\}.$$

For the rest of the proof, the maximum over $\alpha$ is taken within the set $I$. We have

$$\mathcal{A}_n \leq \left\langle\!\!\!\left\langle \sup_{p_t} \mathbb{E}_{y_t \sim p_t} \right\rangle\!\!\!\right\rangle_{t=1}^{n} \left[ \sup_{f \in \Delta, \alpha} \sum_{t=1}^{n} \inf_{\hat{y}_t \in \Delta(\mathcal{F})} \mathbb{E}_{y_t} \left[ \langle \hat{y}_t, y_t \rangle \right] - \mathbb{E}_{e_i \sim f} \left[ \langle e_i, y_t \rangle \right] \right.$$

$$\left. - \frac{\alpha}{2} \left( K \sum_{t=1}^{n} \mathbb{E}_{i \sim f} \left[ \langle e_i, y_t \rangle^2 \right] + K'C(f) \right) - \frac{C(f)}{4\alpha} \right]. \tag{15}$$

Dropping some negative terms, we upper bound the last expression by

$$\left\langle\!\!\!\left\langle \sup_{p_t} \mathbb{E}_{y_t \sim p_t} \right\rangle\!\!\!\right\rangle_{t=1}^{n} \left[ \sup_{f \in \mathcal{F}, \alpha} \sum_{t=1}^{n} \langle f, \mathbb{E}\left[y_t'\right] - y_t \rangle - \frac{K\alpha}{2} \sum_{t=1}^{n} \mathbb{E}_{i \sim f} \left[ \langle e_i, y_t \rangle^2 \right] - \frac{C(f)}{4\alpha} \right].$$

Adding and subtracting $\frac{\alpha}{4} \sum_{t=1}^{n} \mathbb{E}_{y_t'} \left[ \mathbb{E}_{i \sim f} \left[ \langle e_i, y_t' \rangle^2 \right] \right]$,

$$\leq \left\langle\!\!\!\left\langle \sup_{p_t} \mathbb{E}_{y_t \sim p_t} \right\rangle\!\!\!\right\rangle_{t=1}^{n} \left[ \sup_{f \in \mathcal{F}, \alpha} \sum_{t=1}^{n} \langle f, \mathbb{E}\left[y_t'\right] - y_t \rangle - \frac{K\alpha}{4} \sum_{t=1}^{n} \mathbb{E}_{i \sim f} \left[ \langle e_i, y_t \rangle^2 \right] - \frac{K\alpha}{4} \sum_{t=1}^{n} \mathbb{E}_{y_t'} \left[ \mathbb{E}_{i \sim f} \left[ \langle e_i, y_t' \rangle^2 \right] \right] \right.$$

$$\left. + \frac{K\alpha}{4} \left( \sum_{t=1}^{n} \mathbb{E}_{y_t'} \left[ \mathbb{E}_{i \sim f} \left[ \langle e_i, y_t' \rangle^2 \right] \right] - \mathbb{E}_{i \sim f} \left[ \langle e_i, y_t \rangle^2 \right] \right) - \frac{C(f)}{4\alpha} \right].$$

Using Jensen's inequality to pull out expectations, we obtain an upper bound,

$$\left\langle\!\!\left\langle\sup_{p_t}\mathbb{E}_{y_t,y'_t\sim p_t}\right\rangle\!\!\right\rangle_{t=1}^n\left[\sup_{f\in\mathcal{F},\alpha}\sum_{t=1}^n\left\langle f,y'_t-y_t\right\rangle-\frac{K\alpha}{4}\sum_{t=1}^n\mathbb{E}_{i\sim f}\big[\langle e_i,y_t\rangle^2\big]-\frac{K\alpha}{4}\sum_{t=1}^n\mathbb{E}_{i\sim f}\big[\langle e_i,y'_t\rangle^2\big]\right.$$

$$\left.+\frac{K\alpha}{4}\left(\sum_{t=1}^n\mathbb{E}_{i\sim f}\big[\langle e_i,y'_t\rangle^2\big]-\mathbb{E}_{i\sim f}\big[\langle e_i,y_t\rangle^2\big]\right)-\frac{C(f)}{4\alpha}\right].$$

Next, we introduce Rademacher random variables:

$$\left\langle\!\!\left\langle\sup_{p_t}\mathbb{E}_{y_t,y'_t\sim p_t}\mathbb{E}_{\epsilon_t}\right\rangle\!\!\right\rangle_{t=1}^n\left[\sup_{f\in\mathcal{F},\alpha}\sum_{t=1}^n\epsilon_t\left(\langle f,y'_t-y_t\rangle+\frac{K\alpha}{4}\big(\mathbb{E}_{i\sim f}\big[\langle e_i,y'_t\rangle^2\big]-\mathbb{E}_{i\sim f}\big[\langle e_i,y_t\rangle^2\big]\big)\right)\right.$$

$$\left.-\frac{K\alpha}{4}\sum_{t=1}^n\mathbb{E}_{i\sim f}\big[\langle e_i,y_t\rangle^2\big]-\frac{K\alpha}{4}\sum_{t=1}^n\mathbb{E}_{i\sim f}\big[\langle e_i,y'_t\rangle^2\big]-\frac{C(f)}{4\alpha}\right]$$

$$\leq\left\langle\!\!\left\langle\sup_{y_t}\mathbb{E}_{\epsilon_t}\right\rangle\!\!\right\rangle_{t=1}^n\left[\sup_{f\in\mathcal{F},\alpha}\sum_{t=1}^n\epsilon_t\left(2\langle f,y_t\rangle+\frac{K\alpha}{2}\mathbb{E}_{i\sim f}\big[\langle e_i,y_t\rangle^2\big]\right)-\frac{K\alpha}{2}\sum_{t=1}^n\mathbb{E}_{i\sim f}\big[\langle e_i,y_t\rangle^2\big]-\frac{C(f)}{4\alpha}\right].$$

Moving to the tree notation, we have

$$\sup_{\mathbf{y}}\mathbb{E}_\epsilon\sup_{f\in\mathcal{F},\alpha}\left[\sum_{t=1}^n\epsilon_t\left(2\langle f,\mathbf{y}_t(\epsilon)\rangle+\frac{K\alpha}{2}\mathbb{E}_{i\sim f}\big[\langle e_i,\mathbf{y}_t(\epsilon)\rangle^2\big]\right)\right.$$

$$\left.-\frac{K\alpha}{2}\sum_{t=1}^n\mathbb{E}_{i\sim f}\big[\langle e_i,\mathbf{y}_t(\epsilon)\rangle^2\big]-\frac{\mathrm{KL}(f\|\pi)}{4\alpha}-\frac{\log n}{4\alpha}\right].$$

Noting that the convex conjugate of $\frac{1}{\alpha}\mathrm{KL}(f\|\pi)$ is given by $\Psi^*(X)=\frac{1}{\alpha}\log\left(\mathbb{E}_{i\sim\pi}\exp\left(\alpha\langle e_i,X\rangle\right)\right)$, we express the last quantity as

$$\sup_{\mathbf{y}}\mathbb{E}_\epsilon\max_\alpha\frac{1}{4\alpha}\log\left(\mathbb{E}_{i\sim\pi}\exp\left(\sum_{t=1}^n\epsilon_t\big(8\alpha\langle e_i,\mathbf{y}_t(\epsilon)\rangle+2K\alpha^2\langle e_i,\mathbf{y}_t(\epsilon)\rangle^2\big)-2K\alpha^2\langle e_i,\mathbf{y}_t(\epsilon)\rangle^2\right)\right)-\frac{\log n}{4\alpha}.$$

Define a random variable indexed by $\alpha$:

$$X_\alpha=\frac{1}{4\alpha}\log\left(\mathbb{E}_{i\sim\pi}\left[\exp\left(\sum_{t=1}^n\epsilon_t\big(8\alpha\langle e_i,\mathbf{y}_t(\epsilon)\rangle+2K\alpha^2\langle e_i,\mathbf{y}_t(\epsilon)\rangle^2\big)-2K\alpha^2\langle e_i,\mathbf{y}_t(\epsilon)\rangle^2\right)\right]\right).$$

Our goal is to bound $\mathbb{E}\left[\max_\alpha\{X_\alpha-\log(n)/4\alpha\}\right]$. Now notice that

$$P(X_\alpha>t)\leq\inf_\lambda\mathbb{E}\left[e^{\lambda X_\alpha-\lambda t}\right]$$

$$=\inf_\lambda\left\{\mathbb{E}_\epsilon\left(\mathbb{E}_{i\sim\pi}\exp\left(\sum_{t=1}^n\epsilon_t\big(8\alpha\langle e_i,\mathbf{y}_t(\epsilon)\rangle+2K\alpha^2\langle e_i,\mathbf{y}_t(\epsilon)\rangle^2\big)-2K\alpha^2\langle e_i,\mathbf{y}_t(\epsilon)\rangle^2\right)\right)^{\frac{\lambda}{4\alpha}}e^{-\lambda t}\right\}$$

$$\leq\mathbb{E}_\epsilon\mathbb{E}_{i\sim\pi}\exp\left(\sum_{t=1}^n\epsilon_t\big(8\alpha\langle e_i,\mathbf{y}_t(\epsilon)\rangle+2K\alpha^2\langle e_i,\mathbf{y}_t(\epsilon)\rangle^2\big)-2K\alpha^2\langle e_i,\mathbf{y}_t(\epsilon)\rangle^2\right)e^{-4\alpha t}$$

$$\leq\mathbb{E}_\epsilon\mathbb{E}_{i\sim\pi}\exp\left(\sum_{t=1}^n\big(8\alpha\langle e_i,\mathbf{y}_t(\epsilon)\rangle+2K\alpha^2\langle e_i,\mathbf{y}_t(\epsilon)\rangle^2\big)^2-2K\alpha^2\langle e_i,\mathbf{y}_t(\epsilon)\rangle^2\right)e^{-4\alpha t}$$

$$\leq\mathbb{E}_\epsilon\mathbb{E}_{i\sim\pi}\exp\left(\sum_{t=1}^n4\alpha(4+K\alpha)^2\langle e_i,\mathbf{y}_t(\epsilon)\rangle^2-2K\alpha^2\langle e_i,\mathbf{y}_t(\epsilon)\rangle^2\right)e^{-4\alpha t}.$$

The above term is upper bounded by $\exp(-4\alpha t)$ as soon as $4\alpha^2(4+K\alpha)^2\leq 2K\alpha^2$, which happens when

$$0<\alpha\leq(\sqrt{K/2}-4)/K. \tag{16}$$

In view of (14), we know that $\alpha\leq\frac{1}{\sqrt{K'}}$. Thus, to ensure (16), it is sufficient to take $K=50$ and $K'=50^2$. Other choices lead to a different balance of constants. We thus have

$$P(X_\alpha>t)\leq\exp\left(-4\alpha t\right).$$

Now that we have the tail bound, we appeal to Proposition 2. Setting $s_i=4\alpha_i$ and $B_i=1/4\alpha_i$, we obtain that

$$\mathbb{E}\left[\max_{i=1,\ldots,N}\left\{X_{\alpha_i}-\frac{\log(n)}{4\alpha}\right\}\right]\leq 10.$$

$\square$

# B   Relaxations and Algorithms

*Proof of Admissibility for Example 5.1.*

**Lemma 12.** *The following bound is achievable in the setting given in example 5.1:*

$$\mathcal{B}_n(f) = 3\sqrt{2n \max\{KL(f \mid \pi), 1\}} + 4\sqrt{n}. \tag{17}$$

This algorithm can be interpreted as running a "low-level" instance of the exponential weights algorithm for each complexity radius $R_i$, then combining the predictions of these algorithms with a "high-level" instance. The high-level distribution $q_t^\star$ differs slightly from the usual exponential weights distribution in that it incorporates a prior whose weight decreases as the complexity radius increases. The prior distribution prevents the strategy from incurring a penalty that depends on the range of values the complexity radii take on, which would happen if the standard exponential weights distribution were used.

Following the analysis style of Corollary 8, we directly consider an upper bound based on $\mathrm{KL}(f \mid \pi)$ but instead use a complexity-radius-based upper bound with the KL divergence controlling the complexity radius: $\mathcal{F}(R) = \{f : \mathrm{KL}(f \mid \pi) \le R\}$. Concretely, we move from (17) to the bound

$$\mathcal{B}_n(i) = 3\sqrt{nR_i} + 4\sqrt{n}$$

for $R_i = 2^{i-1}$ with $i \in \mathbb{N}$. To keep the analysis as tidy as possible, we will study the achievability of $\mathcal{B}_n(i) = D\sqrt{R_i n}$, setting $D$ and including additive constants only when we reach a point in the analysis where it becomes necessary to do so. The relaxation we consider is

$$\mathbf{Rel}_n(y_{1:t}) = \inf_{\lambda>0}\left[\frac{1}{\lambda}\log\left(\sum_i \exp\left(-\lambda\left[\sum_{s=1}^{t}\langle q_s^{R_i}(y_{1:s-1}), y_s\rangle - 2\sqrt{nR_i} + \mathcal{B}_n(i)\right]\right)\right) + 2\lambda(n-t)\right].$$

**Initial Condition:**   This inequality follows from Lemma 13 and an application of the softmax function as an upper bound on the supremum over $i$:

$$-\inf_i\left[\inf_{f\in\mathcal{F}(R_i)}\sum_{t=1}^{n}\ell(f, y_t) + \mathcal{B}_n(i)\right]$$

$$= \sup_i\left[-\sum_{s=1}^{t}\langle q_s^{R_i}(y_{1:s-1}), y_s\rangle + 2\sqrt{nR_i} - \mathcal{B}_n(i)\right]$$

$$\le \inf_{\lambda>0}\frac{1}{\lambda}\log\left(\sum_i \exp\left(-\lambda\left[\sum_{s=1}^{t}\langle q_s^{R_i}(y_{1:s-1}), y_s\rangle - 2\sqrt{nR_i} + \mathcal{B}_n(i)\right]\right)\right)$$

$$= \mathbf{Rel}_n(y_{1:n}).$$

**Admissibility Condition:**   Define a strategy $q_t^\star$ via

$$(q_t^\star)_i = \frac{\exp\left(-\lambda_t^\star\left[\sum_{s=1}^{t-1}\langle q_s^{R_i}(y_{1:s-1}), y_s\rangle - 2\sqrt{nR_i} + \mathcal{B}_n(i)\right]\right)}{\sum_j \exp\left(-\lambda_t^\star\left[\sum_{s=1}^{t-1}\langle q_s^{R_j}(y_{1:s-1}), y_s\rangle - 2\sqrt{nR_j} + \mathcal{B}_n(R_j)\right]\right)},$$

where we have set

$$\lambda_t^\star = \arg\min_{\lambda>0}\left[\frac{1}{\lambda}\log\left(\sum_i \exp\left(-\lambda\left[\sum_{s=1}^{t-1}\langle q_s^{R_i}(y_{1:s-1}), y_s\rangle - 2\sqrt{nR_i} + \mathcal{B}_n(i)\right]\right)\right) + 2\lambda(n-t+1)\right].$$

We proceed to demonstrate admissibility:

$$\inf_{q_t}\sup_{y_t}\left[\langle q_t, y_t\rangle + \mathbf{Rel}_n(y_{1:t})\right]$$

$$= \inf_{q_t}\sup_{y_t}\left[\langle q_t, y_t\rangle + \inf_{\lambda>0}\left[\frac{1}{\lambda}\log\left(\sum_i \exp\left(-\lambda\left[\sum_{s=1}^{t}\langle q_s^{R_i}(y_{1:s-1}), y_s\rangle - 2\sqrt{nR_i} + \mathcal{B}_n(i)\right]\right)\right) + 2\lambda(n-t)\right]\right].$$

We now plug in $q_t^\star$ and $\lambda_t^\star$ as described above:

$$\le \sup_{y_t}\left[\frac{1}{\lambda_t^\star}\log\left(\exp\left(\lambda_t^\star \mathbb{E}_{i\sim q_t^\star}\langle q_t^{R_i}(y_{1:t-1}), y_t\rangle\right)\right) + \frac{1}{\lambda_t^\star}\log\left(\mathbb{E}_{i\sim q_t^\star}\exp\left(-\lambda_t^\star\langle q_t^{R_i}(y_{1:t-1}), y_t\rangle\right)\right)\right.$$

$$\left. + \frac{1}{\lambda_t^\star}\log\left(\sum_i \exp\left(-\lambda_t^\star\left[\sum_{s=1}^{t-1}\langle q_s^{R_i}(y_{1:s-1}), y_s\rangle - 2\sqrt{nR_i} + \mathcal{B}_n(i)\right]\right)\right) + 2\lambda_t^\star(n-t)\right].$$

We combine the first two terms in the expression and apply Jensen's inequality to arrive at an upper bound:

$$\leq \sup_{y_t}\left[\frac{1}{\lambda_t^\star}\log\Big(\mathbb{E}_{i,i'\sim q_t^\star}\exp\big(\lambda_t^\star\big\langle q_t^{R_i}(y_{1:t-1}) - q_t^{R_{i'}}(y_{1:t-1}), y_t\big\rangle\big)\Big)\right.$$
$$\left. + \frac{1}{\lambda_t^\star}\log\Big(\sum_i\exp\Big(-\lambda_t^\star\Big[\sum_{s=1}^{t-1}\big\langle q_s^{R_i}(y_{1:s-1}), y_s\big\rangle - 2\sqrt{nR_i} + \mathcal{B}_n(i)\Big]\Big)\Big) + 2\lambda_t^\star(n-t)\right].$$

The first term is now bounded using sub-gaussianity.

$$\leq \frac{1}{\lambda_t^\star}\log\Big(\sum_i\exp\Big(-\lambda_t^\star\Big[\sum_{s=1}^{t-1}\big\langle q_s^{R_i}(y_{1:s-1}), y_s\big\rangle - 2\sqrt{nR_i} + \mathcal{B}_n(i)\Big]\Big)\Big) + 2\lambda_t^\star(n-t+1)$$
$$= \inf_{\lambda>0}\left[\frac{1}{\lambda}\log\Big(\sum_i\exp\Big(-\lambda\Big[\sum_{s=1}^{t-1}\big\langle q_s^{R_i}(y_{1:s-1}), y_s\big\rangle - 2\sqrt{nR_i} + \mathcal{B}_n(i)\Big]\Big)\Big) + 2\lambda(n-t+1)\right]$$
$$= \mathbf{Rel}_n(y_{1:t-1}).$$

Having shown that $\mathbf{Rel}_n$ is an admissible relaxation, it remains to show that the relaxation's final value,

$$\mathbf{Rel}_n(\cdot) = \inf_{\lambda>0}\left[\frac{1}{\lambda}\log\Big(\sum_i\exp\big(\lambda[2\sqrt{nR_i} - D\sqrt{nR_i}]\big)\Big) + 2\lambda n\right]$$

is not too large. Setting $D = 3$,

$$= \inf_{\lambda>0}\left[\frac{1}{\lambda}\log\Big(\sum_i\exp\big(-\lambda\sqrt{nR_i}\big)\Big) + 2\lambda n\right].$$

The complexity radius $R_i$ is discretized such that $R_i - R_{i-1} \geq 1$, yielding

$$\leq \inf_{\lambda>0}\left[\frac{1}{\lambda}\log\Big(\exp(-\lambda\sqrt{n}) + \sum_{i=2}^{\infty}(R_i - R_{i-1})\exp\big(-\lambda\sqrt{nR_i}\big)\Big) + 2\lambda n\right]$$
$$\leq \inf_{\lambda>0}\left[\frac{1}{\lambda}\log\Big(\exp(-\lambda\sqrt{n}) + \int_1^\infty\exp\big(-\lambda\sqrt{nR}\big)dR\Big) + 2\lambda n\right].$$

The integral is a routine calculation.

$$\int_1^\infty\exp\big(-\lambda\sqrt{nR}\big)dR = -2\frac{1}{\lambda^2 n}\exp\big(-\lambda\sqrt{nR}\big)\big[\lambda\sqrt{nR} + 1\big]\Big|_1^\infty.$$

Finally, set $\lambda = 1/\sqrt{n}$ yielding

$$\mathbf{Rel}_n(\cdot) \leq 4\sqrt{n}.$$

Note that instead of setting $\lambda_t = \lambda_t^\star$ as described above, we could have set $\lambda_t = 1/\sqrt{n}$ and achieved the same regret bound. $\qquad\square$

**Lemma 13.** *Consider the experts setting from Example 4.5, but with hypothesis class $\mathcal{F}(R) = \{f : KL(f \mid \pi) \leq R\}$. The following inequality holds:*

$$-\inf_{f\in\mathcal{F}(R)}\sum_{t=1}^n\langle y_t, f\rangle \leq -\sum_{t=1}^n\big\langle y_t, q^R(y_{1:t-1})\big\rangle + 2\sqrt{Rn}.$$

*Proof.* Our strategy is to move to an upper bound based on the Kullback-Leibler divergence and exploit convex duality:

$$-\inf_{f\in\mathcal{F}(R)}\sum_{t=1}^n\langle y_t, f\rangle$$
$$\leq -\inf_{f\in\mathcal{F}(R)}\left\{\sum_{t=1}^n\langle y_t, f\rangle + \alpha KL(f \mid \pi)\right\} + \alpha R$$
$$\leq -\inf_{f\in\mathcal{F}}\left\{\sum_{t=1}^n\langle y_t, f\rangle + \alpha KL(f \mid \pi)\right\} + \alpha R.$$

We use $\Psi^\star$ to denote the Fenchel conjugate of $KL(\cdot \mid \pi)$:

$$= \alpha\Psi^\star\left(-\frac{1}{\alpha}\sum_{t=1}^n y_t\right)\Psi + \alpha R.$$

The function $\mathrm{KL}(\cdot \mid \pi)$ is 1-strongly convex, which implies that $\Psi^\star$ is 1-strongly smooth. We peel off one term at a time:

$$\alpha\Psi^\star\left(-\frac{1}{\alpha}\sum_{t=1}^{n} y_t\right) \le \alpha\Psi^\star\left(-\frac{1}{\alpha}\sum_{t=1}^{n-1} y_t\right) + \left\langle -y_n, \nabla\Psi^\star\left(-\frac{1}{\alpha}\sum_{t=1}^{n-1} y_t\right)\right\rangle + \frac{1}{\alpha}.$$

This obtains the following upper bound:

$$-\sum_{t=1}^{n}\left\langle y_t, \nabla\Psi^\star\left(-\frac{1}{\alpha}\sum_{s=1}^{t-1} y_s\right)\right\rangle + \frac{KCn}{\alpha} + \alpha R.$$

Setting $\alpha = \sqrt{n/R}$ and noting that $\nabla\Psi^\star\left(-\sqrt{\frac{R}{n}}\sum_{s=1}^{t-1} y_s\right) = q^R(y_{1:t-1})$ yields the result. $\qquad\square$

***Proof of Lemma 9.*** Recall the form of the $\mathbf{Ada}_n$ relaxation, where we have abbreviated $\mathbf{Rel}_n^R$ to $\mathbf{R}^R$:

$$\mathbf{Ada}_n(y_{1:t}) = \sup_{\mathbf{y},\mathbf{y}'}\mathbb{E}_\epsilon\sup_R\left[\mathbf{R}^R(y_{1:t}) - \mathbf{R}^R\theta(\mathbf{R}^R) + 2\sum_{s=t+1}^{n}\epsilon_s\mathbb{E}_{\hat{y}_s\sim q_s^R(y_{1:t},\mathbf{y}'_{t+1:s-1}(\epsilon))}\ell(\hat{y}_s,\mathbf{y}_s(\epsilon))\right].$$

**Initial Condition:** This directly follows from the fact that $\mathbf{R}^R$ satisfy the initial condition:

$$\mathbf{Ada}_n(y_{1:n}) = \sup_R\left[\mathbf{R}^R(y_{1:n}) - \mathbf{R}^R\theta(\mathbf{R}^R)\right]$$

$$\ge \sup_R\left[-\inf_{f\in\mathcal{F}(R)}\sum_{t=1}^{n}\ell(f,y_t) - \mathbf{R}^R\theta(\mathbf{R}^R)\right]$$

$$= -\inf_R\inf_{f\in\mathcal{F}(R)}\left[\sum_{t=1}^{n}\ell(f,y_t) + \mathbf{R}^R\theta(\mathbf{R}^R)\right].$$

Therefore, playing the strategy corresponding to $\mathbf{Ada}_n$ yields an adaptive regret bound of the form $\mathcal{B}_n(R) = \mathbf{Rel}_n^R(\cdot)\theta(\mathbf{Rel}_n^R(\cdot)) + \mathbf{Ada}_n(\cdot)$.

**Admissibility Condition:** We obtain the following equalities using the same minimax swap technique as in the Lemma 1 proof:

$$\inf_{q_t}\sup_{y_t}\mathbb{E}_{\hat{y}_t\sim q_t}\left[\ell(\hat{y}_t,y_t) + \mathbf{Ada}_n(y_{1:t})\right]$$

$$= \inf_{q_t}\sup_{y_t}\mathbb{E}_{\hat{y}_t\sim q_t}\sup_{\mathbf{y},\mathbf{y}'}\mathbb{E}_\epsilon\sup_R\left[\ell(\hat{y}_t,y_t) + \mathbf{R}^R(y_{1:t}) - \mathbf{R}^R\theta(\mathbf{R}^R)\right.$$

$$\left. + 2\sum_{s=t+1}^{n}\epsilon_s\mathbb{E}_{\hat{y}_s\sim q_s^R(y_{1:t},\mathbf{y}'_{t+1:s-1}(\epsilon))}\ell(\hat{y}_s,\mathbf{y}_s(\epsilon))\right]$$

$$= \sup_{p_t}\mathbb{E}_{y_t\sim p_t}\sup_{\mathbf{y},\mathbf{y}'}\mathbb{E}_\epsilon\sup_R\left[\inf_{\hat{y}_t}\mathbb{E}_{y'_t\sim p_t}\ell(\hat{y}_t,y'_t) + \mathbf{R}^R(y_{1:t}) - \mathbf{R}^R\theta(\mathbf{R}^R)\right.$$

$$\left. + 2\sum_{s=t+1}^{n}\epsilon_s\mathbb{E}_{\hat{y}_s\sim q_s^R(y_{1:t},\mathbf{y}'_{t+1:s-1}(\epsilon))}\ell(\hat{y}_s,\mathbf{y}_s(\epsilon))\right].$$

Note that

$$\inf_{\hat{y}_t}\mathbb{E}_{y'_t\sim p_t}\ell(\hat{y}_t,y'_t) = \inf_{q_t\in\Delta(\mathcal{D})}\mathbb{E}_{\hat{y}_t\sim q_t}\mathbb{E}_{y'_t\sim p_t}\ell(\hat{y}_t,y'_t),$$

and we may replace the infimizing distribution with the randomized strategy $q_t^R$ corresponding to $\mathbf{Rel}_n^R$. The fact that this strategy depends on $y_{1:t-1}$ is left implicit. This yields an upper bound,

$$\sup_{p_t}\mathbb{E}_{y_t\sim p_t}\sup_{\mathbf{y},\mathbf{y}'}\mathbb{E}_\epsilon\sup_R\left[\mathbb{E}_{y'_t\sim p_t}\mathbb{E}_{\hat{y}_t\sim q_t^R}\ell(\hat{y}_t,y'_t) + \mathbf{R}^R(y_{1:t}) - \mathbf{R}^R\theta(\mathbf{R}^R)\right.$$

$$\left. + 2\sum_{s=t+1}^{n}\epsilon_s\mathbb{E}_{\hat{y}_s\sim q_s^R(y_{1:t},\mathbf{y}'_{t+1:s-1}(\epsilon))}\ell(\hat{y}_s,\mathbf{y}_s(\epsilon))\right],$$

which we can write by adding and subtracting $\mathbb{E}_{\hat{y}_t\sim q_t^R}\ell(\hat{y}_t,y_t)$ as

$$\sup_{p_t}\mathbb{E}_{y_t\sim p_t}\sup_{\mathbf{y},\mathbf{y}'}\mathbb{E}_\epsilon\sup_R\left[\mathbb{E}_{y'_t\sim p_t}\mathbb{E}_{\hat{y}_t\sim q_t^R}\ell(\hat{y}_t,y'_t) - \mathbb{E}_{\hat{y}_t\sim q_t^R}\ell(\hat{y}_t,y_t) + \mathbb{E}_{\hat{y}_t\sim q_t^R}\ell(\hat{y}_t,y_t)\right.$$

$$\left. + \mathbf{R}^R(y_{1:t}) - \mathbf{R}^R\theta(\mathbf{R}^R) + 2\sum_{s=t+1}^{n}\epsilon_s\mathbb{E}_{\hat{y}_s\sim q_s^R(y_{1:t},\mathbf{y}'_{t+1:s-1}(\epsilon))}\ell(\hat{y}_s,\mathbf{y}_s(\epsilon))\right].$$

Now, using the fact that $\mathbf{R}^R$ are admissible,

$$\leq \sup_{p_t} \mathbb{E}_{y_t \sim p_t} \sup_{\mathbf{y}, \mathbf{y}'} \mathbb{E}_\epsilon \sup_R \left[ \mathbb{E}_{y'_t \sim p_t} \mathbb{E}_{\hat{y}_t \sim q_t^R} \ell(\hat{y}_t, y'_t) - \mathbb{E}_{\hat{y}_t \sim q_t^R} \ell(\hat{y}_t, y_t) \right.$$
$$\left. + \mathbf{R}^R(y_{1:t-1}) - \mathbf{R}^R \theta(\mathbf{R}^R) + 2 \sum_{s=t+1}^n \epsilon_s \mathbb{E}_{\hat{y}_s \sim q_s^R(y_{1:t}, \mathbf{y}'_{t+1:s-1}(\epsilon))} \ell(\hat{y}_s, \mathbf{y}_s(\epsilon)) \right].$$

By Jensen's inequality, we upper bound the last expression by

$$\sup_{p_t} \mathbb{E}_{y_t, y'_t \sim p_t} \sup_{\mathbf{y}, \mathbf{y}'} \mathbb{E}_\epsilon \sup_R \left[ \mathbb{E}_{\hat{y}_t \sim q_t^R} \ell(\hat{y}_t, y'_t) - \mathbb{E}_{\hat{y}_t \sim q_t^R} \ell(\hat{y}_t, y_t) + \mathbf{R}^R(y_{1:t-1}) - \mathbf{R}^R \theta(\mathbf{R}^R) \right.$$
$$\left. + 2 \sum_{s=t+1}^n \epsilon_s \mathbb{E}_{\hat{y}_s \sim q_s^R(y_{1:t}, \mathbf{y}'_{t+1:s-1}(\epsilon))} \ell(\hat{y}_s, \mathbf{y}_s(\epsilon)) \right].$$

We now replace each choice $y_t$ in the last sum by a worst-case choice $y''_t$:

$$\leq \sup_{p_t} \mathbb{E}_{y_t, y'_t \sim p_t} \sup_{y''_t} \sup_{\mathbf{y}, \mathbf{y}'} \mathbb{E}_\epsilon \sup_R \left[ \mathbb{E}_{\hat{y}_t \sim q_t^R} \ell(\hat{y}_t, y'_t) - \mathbb{E}_{\hat{y}_t \sim q_t^R} \ell(\hat{y}_t, y_t) + \mathbf{R}^R(y_{1:t-1}) - \mathbf{R}^R \theta(\mathbf{R}^R) \right.$$
$$\left. + 2 \sum_{s=t+1}^n \epsilon_s \mathbb{E}_{\hat{y}_s \sim q_s^R(y_{1:t-1}, y''_t, \mathbf{y}'_{t+1:s-1}(\epsilon))} \ell(\hat{y}_s, \mathbf{y}_s(\epsilon)) \right].$$

We then introduce $\epsilon_t$ since $y_t, y'_t$ can be renamed. The last expression is equal to

$$\sup_{p_t} \mathbb{E}_{y_t, y'_t \sim p_t} \mathbb{E}_{\epsilon_t} \sup_{y''_t} \sup_{\mathbf{y}, \mathbf{y}'} \mathbb{E}_\epsilon \sup_R \left[ \mathbb{E}_{\hat{y}_t \sim q_t^R} \left[ \epsilon_t (\ell(\hat{y}_t, y'_t) - \ell(\hat{y}_t, y_t)) \right] + \mathbf{R}^R(y_{1:t-1}) - \mathbf{R}^R \theta(\mathbf{R}^R) \right.$$
$$\left. + 2 \sum_{s=t+1}^n \epsilon_s \mathbb{E}_{\hat{y}_s \sim q_s^R(y_{1:t-1}, y''_t, \mathbf{y}'_{t+1:s-1}(\epsilon))} \ell(\hat{y}_s, \mathbf{y}_s(\epsilon)) \right].$$

By splitting into two terms we arrive at an upper bound of

$$\sup_{p_t} \mathbb{E}_{y_t \sim p_t} \mathbb{E}_{\epsilon_t} \sup_{y''_t} \sup_{\mathbf{y}, \mathbf{y}'} \mathbb{E}_\epsilon \sup_R \left[ 2\epsilon_t \mathbb{E}_{\hat{y}_t \sim q_t^R} \left[ \ell(\hat{y}_t, y_t) \right] + \mathbf{R}^R(y_{1:t-1}) - \mathbf{R}^R \theta(\mathbf{R}^R) \right.$$
$$\left. + 2 \sum_{s=t+1}^n \epsilon_s \mathbb{E}_{\hat{y}_s \sim q_s^R(y_{1:t-1}, y''_t, \mathbf{y}'_{t+1:s-1}(\epsilon))} \ell(\hat{y}_s, \mathbf{y}_s(\epsilon)) \right]$$
$$= \sup_{y_t} \mathbb{E}_{\epsilon_t} \sup_{y''_t} \sup_{\mathbf{y}, \mathbf{y}'} \mathbb{E}_\epsilon \sup_R \left[ 2\epsilon_t \mathbb{E}_{\hat{y}_t \sim q_t^R} \left[ \ell(\hat{y}_t, y_t) \right] + \mathbf{R}^R(y_{1:t-1}) - \mathbf{R}^R \theta(\mathbf{R}^R) \right.$$
$$\left. + 2 \sum_{s=t+1}^n \epsilon_s \mathbb{E}_{\hat{y}_s \sim q_s^R(y_{1:t-1}, y''_t, \mathbf{y}'_{t+1:s-1}(\epsilon))} \ell(\hat{y}_s, \mathbf{y}_s(\epsilon)) \right]$$
$$= \mathbf{Ada}_n(y_{1:t-1}).$$

$\square$