[Reviews · NeurIPS 2015]

Submitted by Assigned_Reviewer_1

Summary:

This paper investigates luckiness guarantees for online learning that may depend on the data and on the comparator. Deciding achievability of a proposed adaptive bound is cast as a minimax problem. This problem is subsequently relaxed using sequential Rademacher complexity. Probability tail bounds are used to give tractable expressions for achievable bounds in general and in a variety of examples.

Originality and Significance:

This is a novel extension of the game-theoretic approach to obtain adaptive bounds. Whereas most algorithms are designed bottom-up, this approach starts at the core of the problem and. Tractability then becomes a trade-off with sharpness of bounds. This method is very powerful, and will undoubtedly result in new algorithms for existing and new adaptivity targets.

Clarity and Quality:

The paper is nicely done. I was a little disappointed that the methods, after plugging in the proposed tail bounds, unfortunately cannot match the state of the art in the worked-out examples. I also feel that the discussion of the results is a little one-sided as it does not point this out enough.

The bound of Example 4.4 should be compared to Theorem 1 and Corollary 1 from the NIPS'14 paper "Exploiting Easy Data in Online Optimization" by Sani, Neu and Lazaric. Their bound seems strictly better, as it gives constant regret compared to the singled-out strategy, independent of the time horizon and the number of other strategies. I am also a little wary of the current bound's dependence on $L$. Intuitively the whole bound should simply scale with $L$, for this is what should happen if we rescale the loss function. (if not, the results can be improved by pre- and post-scaling appropriately).

The bound of Example 4.5 should be compared to Theorem 4 (which trivially extends from subsets to mixtures) from the COLT'15 paper "Second-order Quantile Methods for Experts and Combinatorial Games" by Koolen and Van Erven. Doing so, the advantage that the current paper sees over [13] (which is obtained by magnifying lower-order terms) disappears. What remains is that the current paper has a worse linear term (loss of comparator f instead of variance of regret with respect to f) at the cost of a worse multiplicative overhead (ln n at time horizon n instead of ln ln n). It is surprising that the proposed method, which in principle should be much tighter as it is minimax-optimal, ignores all computational issues and fixed-horizon, loses out on all fronts to a computationally efficient anytime method. It would be interesting to see a dissection of where the advantage is lost.

Minor:

line 161: move the left-hand-side of this definition from line 160 into the equation.

lines 272, 372: You need a singleton $\cal X$. If it is empty everything collapses.

line 297: Why use O-tilde if you spell out the log factor?

line 304: It looks that empty $\cal F(1)$ presents problems. For what is the Rademacher complexity of the empty set? If it is $0$ we have a division by $0$. If it is $-\infty$ the bound 304 is unachievable.

line 311: Why is infinite set of radii an issue for exponential weights?

line 336: omit "is"

Example 4.4. I was unsure what $N$ means here. Is $N = |\cal F|$?

line 418: Perhaps omit this, it sounds a little embarrassing. You did find the appendix for other material.

page 9: please unify the reference style (full names vs initials) and check for capitals in titles: Hilbert, Banach, ...

*** UPDATE *** I would like to respond to this sentence from the author response:

The bound of Koolen et al is N-independent, but is in terms of the prior (that is, MDL-style).

The suggested difference is purely cosmetic and not intrinsic. A trivial modification of the analysis results in a KL (i.e. PAC-Bayes) bound. Koolen and Van Erven seem to do this for their Theorem 8 but not their Theorem 4. Yet, there is no difficulty whatsoever.
Summary: The paper applies game-theoretic methods to obtaining adaptive regret guarantees. The techniques are very general and powerful, but unfortunately undershoot the state of the art in the worked-out examples.

Submitted by Assigned_Reviewer_2

Over the last few years, Rakhlin, Shridharan and collaborators have built up a learning theory for online learning based on sequential Rademacher complexities, which can characterize the achievable minimax rates for many function classes. So far, however, this theory has not been able to capture adaptive rates, which are non-uniform over the model and over "nice" data sequences. This is the contribution of the present paper: it extends the sequential Rademacher complexities framework to prove adaptive rates. Like for the non-adaptive case, the theory is non-constructive: it does not immediately give any efficient algorithms to achieve the rates.

Technically the new results are obtained by using one-sided concentration inequalities to control an empirical process that involves the Rademacher complexities and the adaptive rate of interest.

Significance

Since adaptive rates are recently receiving significant attention, a better understanding of which adaptive rates are possible is an important contribution. A strong point of this work is that it obtains a very diverse selection of bounds.

In this regard, I believe that Example 4.1 (about the spectral norm) and Corollary 7 (an abstract result for model selection) are new. Example 4.2 recovers a bound by [15], and Example 4.3 and 4.4 are in the spirit of results by [9], but go further.

Example 4.5 is also definitely interesting, but the paper's discussion of its relation to the literature is not quite clear to me. In particular, it is claimed that this example subsumes and improves upon the results of [5], but the term sum_{t=1}^n \langle f,y_t \rangle in the bound is a small-loss/L*-type bound, which is different (and probably weaker) than the cumulative magnitude C_T of instantaneous regrets used in [5]. The dependence on n is also worse in Example 4.5 compared to [5], where it is O(log log n).

Presentation and correctness

The paper is well-written, but the proofs sometimes jump over some details that might be important:

Lemma 1:

* the proof says that additional mild assumptions (like compactness)

are required to swap the min and the max, so please state these in

the lemma.

* The proof of (5) assumes that swapping a derivative and an

expectation is allowed. Please provide a reference or state as an

assumption.

Proposition 2:

* In line 780 it seems that you choose delta_i = B_i(theta_i - 1).

If not, line 780 is not clear to me. If so, please state this

explicitly.

* You exclude delta from the final bound based on the reasoning that

it can be chosen arbitrarily small. But this is not possible if you

fix delta_i = B_i(theta_i - 1). And if your reasoning is correct after all, why

don't you simply set delta_i=0 from the start?

Minor comments:

* Throughout the paper, you handle the settings where x is absent by

setting cal{X} = emptyset. But, formally speaking, nature cannot

choose from an empty set, and I don't know how functions from the

empty set to the reals are defined. So perhaps you should set cal{X}

to be a singleton set instead.

* In the displays in section 1.1, I think you should add expectations

of hat{y}_t under q_t.

* In section 1.2: quantile bounds also depend on the data, because

the epsilon-quantile is the set of experts with cumulative loss

within epsilon of the best expert.

* line 221: please also define this for p=infty, because that's what

you use it for.

* Lemma 4: R_n(G) -> 0 is not possible, because you did not divide by

n in its definition in this paper.

* Lemma 5: N(G,delta) is for which p?

* Corollary 6: "small constant" is not a formal statement, so cannot

be part of a corollary.

* line 594 has a typo in the argmin

* line 611: say you are scaling the difference of derivatives by

1/(2L) to obtain s_t in [-1,+1].

Summary: Since adaptive rates are recently receiving significant attention, a better understanding of which adaptive rates are possible is an important contribution. A strong point of this work is that it obtains a very diverse selection of bounds.

Submitted by Assigned_Reviewer_3

The paper considers the problem of deriving adaptive regret bound in online learning framework, which can be categorized into three types: 1) depends only on f, 2) only on data, and 3) on both quantiles.

As the paper claimed, the paper proposes a general framework which recovers the vast majority of known adaptive rates in literatures, including variance bounds, quantile bounds, localization-based bounds, and fast rates for small losses.

The paper is well-written and clearly stated the main points.

The paper is a theoretical paper, which is beyond the scope of the reviewer's expertise.

The reviewer is concerned with the following issues:

1) Originality of the claimed results:

What specific tools are proposed and devised to derive the adaptive regret bounds are blur to the reviewer, which makes the reviewer conjecture some results are reinvented the wheels by using the existing results of concentration inequality, the sequential Rademacher complexity, and the offset processes.

2) Significance: How the theoretical results connect with concrete online learning algorithms is unknown.

The paper seems assuming a general oracle algorithm exists, then derives the corresponding adaptive regret bound.

How the theoretical results guide concrete online learning algorithms is unknown.

Given the bounds, what can be improved is unknown.

3) No empirical evaluation makes it hard to estimate the correctness of the bounds and give some visual results.

Minor comments: 1. In pp. 2, line 63, "the the right"

2. In pp. 2, line 074, X=\emptyset seems not correct as x_t belongs to X.

Summary: The paper is a theoretical paper to propose a general framework for adaptive regret bounds, including model-selection bounds and data-dependent bounds, in the online learning framework.

No empirical evaluation is presented.

Author Feedback
Author rebuttal: We thank all the reviewers for the useful comments and suggestions.

Let us address the following main issues raised by the reviewers:
1. correct scaling in Examples 4.4 and 4.5 (Reviewers 1 and 2)
2. computationally efficient methods (Reviewers 4 and 6)
3. originality, significance (Reviewer 3)

Regarding issue #1, it is indeed the case that we did not match the scaling in some of the regimes in examples 4.4 and 4.5. This stems from the fact that the main result is very general (Corollary 6 holds well beyond finite or countably infinite F), and we were not careful about the logarithmic factors for ``small classes''. We went back to the proofs and were able to remove these extraneous terms (we will make the modification in the final draft) in Example 4.4.

More specifically:

a. ``The bound of Example 4.4 should be compared to Theorem 1 and Corollary 1 from the NIPS'14 paper ``Exploiting Easy Data in Online Optimization'' by Sani, Neu and Lazaric ...'': Thank you for pointing out the reference, we were unaware of this work. A simple modification of Corollary 6 (specifically catered to finite function class case) yields constant regret against f* and a O(sqrt(n log N) (log n + log log N) ) against arbitrary benchmarks.

b. ``The bound of Example 4.5 should be compared to Theorem 4 (which trivially extends from subsets to mixtures) from the COLT'15 paper ``Second-order Quantile Methods for Experts and Combinatorial Games'' by Koolen and Van Erven''. Thank you for the reference -- we only learned about this paper after the submission. Let us mention a few points of comparison of our work with regard to this paper, as well as [13] by Luo and Schapire. The bound of Koolen et al is N-independent, but is in terms of the prior (that is, MDL-style). The bound of [13] is in terms of KL (as in PAC-Bayes), yet has N dependence. Note that the bound of Example 4.5 has no dependence on N and depends on the prior through KL, so it is a bona fide PAC-Bayesian bound; as such, it can be used for uncountably infinite classes. Second, both 4.4 and 4.5 are algorithm-independent regret bound.

We also remark that in Example 4.5 (specifically, the proof of Lemma 9), we obtained the L* bound starting from a quadratic variation term. Keeping this quadratic term resolves, at least non-constructively, the open question of Cesa-Bianchi, Mansour, Stoltz '06 about an algorithm-independent variance bound for experts. Example 4.3, however, goes well beyond this open question and gives a spectrum of variance bounds for rich classes.

In sum, the extraneous log factors appeared in our original submission because the main result was catered to work well beyond the finite case (this finite case is the focus of the references provided by the reviewers).

Regarding issue #2, it is indeed the case that our results are non-constructive. However, in the last part we outline a generic recipe for obtaining algorithms. The original work on sequential complexities was also non-constructive, but the understanding of correct complexities led to many novel algorithms in subsequent papers. The development of computationally efficient adaptive methods is surely an exciting area of further investigation.

Regarding issue #3, with respect to the novelty, we respectfully disagree: we introduce online PAC bayes theorems, provide variance bounds well beyond the finite class, address an open question by Cesa-Bianchi et al, and, more importantly, show that all of these follow with a unified analysis. Our work lays out a unifying approach for theoretical analysis of adaptive online learnability which we term ``achievability''.